# Benchmarking EGF signaling pathway inference using phosphoproteomics and kinase-substrate interactions

Martin Garrido-Rodriguez [1,2,3,4,5], Clement Potel [1,5], Mira Lea Burtscher [1], Isabelle Becher [1], Pablo Rodriguez-Mier [2,3], Sophia Müller-Dott [2,3], Mikhail M. Savitski [1] ✉ & Julio Saez-Rodriguez [2,3,4] ✉

Signaling pathways are useful models for interpreting molecular data, but their coverage has long been constrained by classic biochemistry methods. The growing corpus of kinase-substrate interactions, coupled to phosphoproteomics improvements, pave the way to revisit classic signaling pathways. In this study, we explore context-specific signaling pathway inference from phosphoproteomics and kinase-substrate networks. Focusing on epidermal growth factor (EGF), we conduct a meta-analysis and generate three datasets representing the most comprehensive characterization of the EGF response to date. We infer kinase-kinase pathways and compare them to different ground truth sets. Literature-curated networks consistently yield the highest recovery of ground-truth interactions, with modest gains from network propagation methods. Up to 90% of interactions are absent from current ground truth sets, indicating many unexplored interactions supported by data and knowledge. Our results demonstrate the limitations of traditional views on signaling pathways and point to opportunities for generating better mechanistic hypotheses.

Cells sense and adapt to internal and external stimuli through signaling processes[1]. Proteins play a central role in signaling, acting as key mediators that transmit information through various mechanisms, including protein-protein interactions, post-translational modifications, targeted degradation, or subcellular translocation.

Among these mechanisms, reversible protein phosphorylation stands out as a highly dynamic process, functioning as a molecular switch that rapidly modulates protein activity and interactions without the need for transcriptional regulation. It plays an essential role in signal transduction through phosphorylation cascades linking external stimuli to gene regulation, enabling cellular adaptation to changing conditions. Dysregulation of phosphorylation-driven signaling has been implicated in numerous diseases, including cancer[2,3],

neurodegenerative disorders[4], and autoimmune diseases[5], highlighting the importance of understanding these mechanisms for advancing both basic and clinical research.

In biochemistry, the term "signaling pathway" refers to conceptual models that represent the flow of information within cells, often depicted as linear molecular cascades[6]. These pathways, typically involving dozens of proteins, have been used to understand how breast cancer cells respond to drug treatment[7], or to determine which rewiring mechanisms drive drug resistance in melanoma cells[8]. However, one of the key limitations of current signaling pathways lies in the restricted scope of traditional biochemical techniques. For many years, low-throughput, antibody-based assays have been used to study signaling processes, leading to constrained pathways[9]. Metaphorically,

[1]Molecular Systems Biology unit, European Molecular Biology Laboratory (EMBL), Heidelberg, Germany. [2]Heidelberg University, Faculty of Medicine, Heidelberg University Hospital, Heidelberg, Germany. [3]Institute for Computational Biomedicine, Heidelberg University Hospital, Heidelberg, Germany. [4]European Molecular Biology Laboratory, European Bioinformatics Institute (EMBL-EBI), Hinxton, Cambridgeshire, UK. [5]These authors contributed equally: Martin Garrido-Rodriguez, Clement Potel. ✉e-mail: mikhail.savitski@embl.de; saezlab@ebi.ac.uk

we have only searched for answers under the streetlight, limiting our view of the broader landscape of signaling interactions. Recent advances in mass spectrometry-based phosphoproteomics, however, have enabled the identification and quantification of tens of thousands of phosphosites in a single experiment[10–12]. To deepen our understanding of these sites and the kinases responsible for their phosphorylation, several approaches have emerged, ranging from computational methods to assess the functional relevance of phosphosites[13] and predict kinase targets using protein language models[14], to the expansion of the kinase–substrate interactions using peptide-array screenings[15,16].

While these in silico and in vitro approaches have provided unprecedented insights into the "dark phosphoproteome"[17], how to use this vast amount of information to uncover previously hidden signaling cascades remains a major challenge. Building more comprehensive pathways requires not only identifying active kinases in a given context, but also understanding how they work together. Previous studies have evaluated signaling network reconstruction leveraging both correlation across studies and information from sequence and structural analysis[18,19]. However, it has not yet been explored to what degree signaling pathways learned from using context-specific data recapitulate and/or expand well-characterized signaling pathways. Such an analysis would make it possible to directly compare established, textbook descriptions of signaling pathways with the activation patterns uncovered by modern high-coverage phosphoproteomic methods. This comparison would not only quantify how well current context-specific data support traditional pathway models but also reveal additional connections or alternative signaling routes that extend or refine our understanding. Moreover, it would help differentiate the contributions of context-specific data and the choice of prior-knowledge resources.

In this study, we adopt the perspective of a naive researcher with the goal of learning a signaling pathway, starting from differential phosphoproteomic data and a kinase–substrate network. We start by combining multiple kinase–substrate networks using literature curation, protein language models, and data from peptide-array screenings. Focusing on the epidermal growth factor (EGF) response, central to diseases such as melanoma, breast and lung cancer, and glioblastoma, we perform a meta-analysis of recent studies and complement them with three deep, time-resolved phosphoproteomic experiments. We also include two recent datasets in which a stimulus different from EGF was used. Using these datasets and networks, we generate context-specific signaling pathways through various computational approaches and compare the results against several ground truth interaction sets. Our findings provide guidance on the effectiveness of different resources and algorithms. Only a small fraction of selected interactions overlap with the ground truth sets, highlighting the vast space of kinase–kinase interactions that have yet to be discovered, and how these can substantially expand our current view of signaling pathways and druggable targets.

## Results

### Combining state-of-the-art kinase–substrate networks

A key challenge in phosphoproteomics is that kinase–substrate networks mapping interactions between a kinase and its substrates currently cover a small fraction of measured human phosphosites (less than 5%[17]). To address this limitation, the first step in our study focused on gathering kinase–substrate networks built under different methodological principles. We began by collecting literature-supported interactions from OmniPath[20] (see "Methods" section). This represents a 'classic' resource on which every interaction is supported by literature evidence from low-throughput experiments. As a state-of-the-art computational predictive method, we used Phosformer, a method that leverages protein language models to infer kinase substrates from sequence[14]. This approach allowed us to generalize known interactions

from literature to similar kinases and substrates. Finally, we also incorporated interactions from the recently published kinase libraries, which used peptide arrays to determine in vitro kinase affinity for specific phospho-motifs[15,16]. Unlike the other two approaches, this resource does not rely on literature data and instead screens and weights all kinases equally. By combining these resources, we generated a more comprehensive kinase–substrate network that spans a significantly larger portion of the phosphoproteome.

To determine appropriate thresholds for the two predictive resources (Phosformer and the kinase libraries), we leveraged data from Hijazi et al.[21], who analyzed the phosphoproteomic response of two cancer cell lines to 60 kinase inhibitors and conducted complementary in vitro affinity assays to identify their target kinases (Fig. 1a). After obtaining the differential phosphoproteomic profiles from the kinase inhibitor treatments, we analyzed kinase activity by applying the Z-score method described in Müller-Dot et al.[22] (Fig. 1b). This analysis used the three previously described resources, plus a combined resource containing the union of interactions from all three. We applied three thresholds for each resource (lenient, moderate, and strict), retrieving all interactions above these thresholds (see "Methods" section). For each resulting network (defined by a specific resource and threshold), we calculated the proportion of phosphosites covered (Coverage) and the AUROC for kinase activities, using drug targets from the complementary in vitro assay as true positives (Fig. 1a; see "Methods" section).

When we analyzed kinases covered by all resources, every expanded network achieved a higher AUROC than the literature-based network. Although differences in predictive performance between resources were modest, the best score came from Phosformer under the lenient threshold (AUROC = 0.67, Fig. 1c, Supplementary Fig. 1a). Data coverage ranged from an average of 5% in the literature-based network to 88% in the combined lenient network. Similar results were obtained when changing the inhibition cutoff that we used to consider a kinase a drug-target (Supplementary Fig. 1b). Although the coverage gain from the lenient to the moderate threshold was small, the lenient threshold nearly doubled the number of interactions compared to the moderate threshold (interaction ratios: Phosformer = 3.61, kinase libraries = 1.97, combined = 2.52). Because the larger number of interactions could increase the risk of false positives, we selected the moderate threshold for further analyses, considering it a reasonable compromise between introducing interactions and maintaining coverage and accuracy.

In a descriptive analysis of the expanded networks, we observed that the numbers of phosphosites, phosphoproteins, and interactions were far greater than those reported in the literature (Fig. 1d). We next examined how these interactions were distributed across major kinase families for each resource (Fig. 1e). In the literature-based network and kinase libraries, most interactions involved Serine/Threonine kinases (82% and 93%, respectively), whereas Phosformer had a higher proportion of interactions for Tyrosine kinases (62%). Dual-specificity kinases were underrepresented in all resources, comprising less than 5% of interactions. Phosformer's greater coverage of Tyrosine kinases may result from biases in its training data or from a better performance of the language model in this kinase subfamily.

We also examined the overlap for kinases with similar sequences. To do so, we calculated the overlap coefficient for target sites among kinases with >50% sequence similarity ($N = 1677$) (Fig. 1f). This analysis revealed that kinases with similar sequences share a higher proportion of target sites in Phosformer (81%) compared to the kinase libraries (60%) or literature (26%), consistent with Phosformer's reliance on kinase sequences as input. We also explored specialized sites (targeted by ≤3 kinases) and convergent sites (targeted by ≥3 kinases). Inversely related to coverage differences, literature-based network had the highest proportion of specialized sites (95%), followed by Phosformer (40%) and the kinase library (23%) (Fig. 1g). Overall, this resource

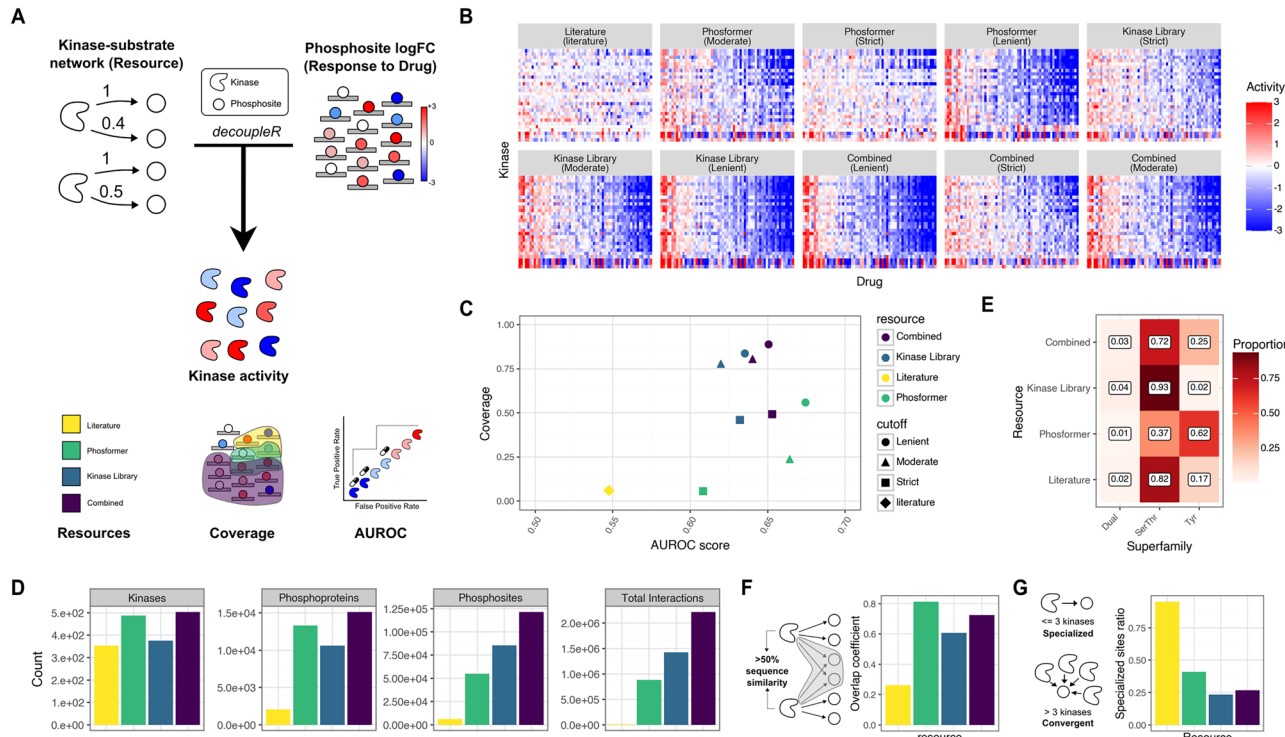

**Fig. 1 | Overview of expanded kinase–substrate networks. A** Schematic of the analysis used to assess coverage and accuracy of different resources, based on data from Hijazi et al. **B** Heatmaps showing estimated activity for common kinases (Y-axis) across drug-response experiments (X-axis) for each resource (facet). **C** Evaluation of resources using kinase activities, where the Y-axis indicates the proportion of phosphosites covered by at least one upstream kinase and the X-axis shows the AUROC for estimated kinase activities, restricted to kinases present in all resources; point shapes represent different thresholds for expanded networks. AUROC scores were estimated through an iterative sampling (100 iterations) with a 1:1 ratio of true positives (inhibited kinases) to true negatives (non-inhibited

kinases). **D** Bar plots showing the number of kinases, phosphoproteins (proteins with at least one phosphorylation site), phosphosites, and interactions for each resource using the moderate threshold. **E** Heatmap showing the proportion of interactions assigned to kinases from different superfamilies for each resource using the moderate threshold. **F** Bar plots of the Szymkiewicz–Simpson coefficient (overlap coefficient) for target sites shared between kinases with high sequence similarity (>50%) across resources using the moderate threshold ($N = 1677$). **G** Bar plots showing the proportion of specialized sites (targeted by ≤3 kinases) per resource using the moderate threshold.

integrates recent computational and experimental methods to greatly expand the number of kinase–substrate interactions. The four context-agnostic resources used in this study are accessible via a Python interface (see "Methods" section), allowing researchers to easily examine how different prior-knowledge networks influence downstream analyses.

## Comprehensive meta-analysis of the EGF signaling response via deep phosphoproteomics

We then focused on gathering phosphoproteomics data characterizing a well-studied pathway: the signaling response to epidermal growth factor (EGF). We chose this pathway for several reasons. First, it is one of the most thoroughly researched signaling cascades, with well-documented and validated kinase–kinase interactions. Second, similar to other ligand-induced processes, it is characterized by a rapid response, with activation occurring within seconds of EGF stimulation[23]. This prevents other phenomena, such as transcriptional regulation, from having a pronounced effect if data is collected early enough. Third, it plays a central role in diseases like cancer, with many proteins in this pathway—such as EGFR, BRAF, and AKT/mTOR—historically serving as key targets for drug development efforts. Lastly, since EGF stimulation is commonly used as a standard experimental perturbation in phosphoproteomics, there is a wealth of public data available.

We began by conducting a meta-analysis of recently published data (see "Methods" section for selection criteria). We selected three studies for our analysis: Skowronek et al.[11], Bortel et al.[10], and Lancaster

et al.[24]. In each study, HeLa cells were stimulated with EGF for a short time, and changes at the phosphoproteome level were quantified on a single time point using different methods (10 min in Skowronek et al., 8 min in Bortel et al., and 15 min in Lancaster et al.). For each study, we retrieved processed data and performed a differential-abundance analysis comparing EGF-treated samples with their respective controls. These studies provided the baseline for state-of-the-art phosphoproteomics analysis of EGF response.

To complement these studies, we also generated three independent datasets in a different biological context. First, we stimulated human embryonic kidney cells (HEK293T) with EGF and quantified the phosphoproteome response at 3, 9, and 25 min, representing very early, early, and late signaling phases based on insights from the literature[7]. To further explore the dynamics of EGF signaling, assuming uneven ligand distribution on attached cells, we aimed to use suspension cells. We conducted a second experiment using HEK293F cells, the suspension-adapted variant of HEK293T, investigating the same time points. Finally, we used HEK293F cells again, but this time measured the phosphoproteomic response at 1, 2, 3, 5, 7, 9, 12, and 15 min of EGF stimulation. To control for cell-line and stimulus specificity, we also incorporated two additional datasets in the meta-analysis. The first, from Chen et al.[25], examined the response of HEK293T cells to a different stimulus, Interferon-alpha (10 min after stimulation). The second, from Tuechler et al., investigated a distinct biological context—the response of kidney cells expressing PDGFR-beta to transforming growth factor-beta[26].

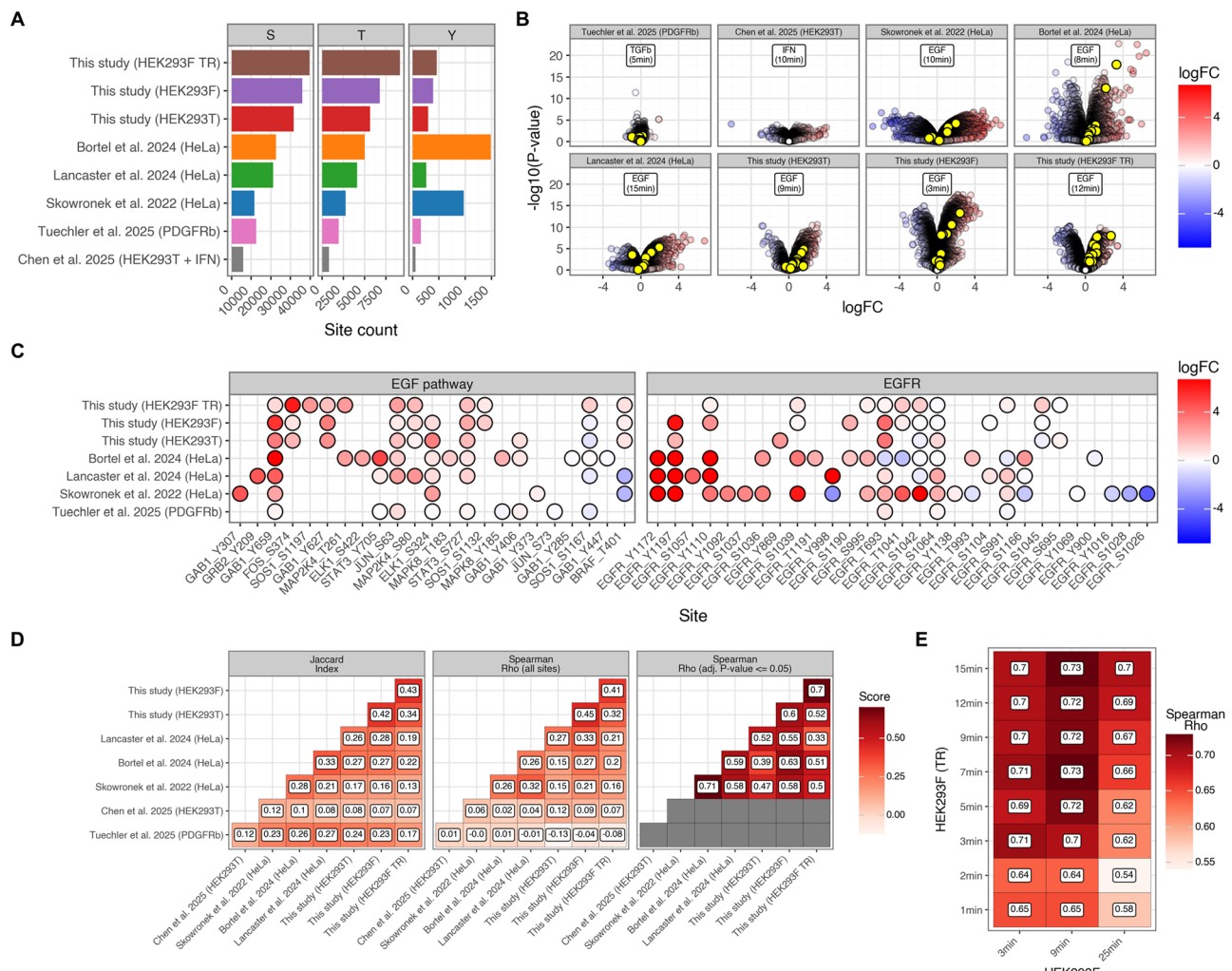

**Fig. 2 | Phosphosite-level overview of studies included in the meta-analysis.**
**A** Bar plots showing the number of identified sites per study, separated by residue type (S serine, T threonine, Y tyrosine). **B** Volcano plots displaying changes in magnitude (x-axis: logFC) and significance (y-axis: −log10 adjusted P-value) for each study. The moderated T-statistic from limma was employed to obtain the differential-abundance estimates. Labels indicate the stimulus and time point. Phosphosites belonging to the SIGNOR EGFR signaling pathway are highlighted in yellow and shown in more detail in the next panel. **C** Site-level view of phosphosites within SIGNOR EGFR pathway (left) and from the EGF receptor (EGFR, right). Colors represent logFC values. **D** Heatmaps comparing similarity between studies. The left panel shows Jaccard indices (intersection/union of identified sites), while the middle and right panels show Spearman correlation coefficients of log fold-changes across all sites and across significant sites only (adjusted P < 0.05). **E** Heatmap of Spearman correlation coefficients for phosphosite log fold-changes at different post-EGF stimulation time points in the two HEK293F experiments.

The three datasets generated in this study provide the most extensive quantitative coverage of the EGF response to date, with a total of 37,620, 43,173, and 49,343 high-confidence phosphosites (localization probability ≥ 0.75) for the HEK293T, HEK293F, and HEK293F time-resolved (TR) datasets, respectively, while the other studies were all below 30,000. Looking at specific phosphorylated residues, our data showed deepest coverage for serines and threonines, while Bortel and Skowronek datasets showed a higher number of identified tyrosines (Fig. 2a).

To explore whether EGF stimulation produces consistent complex temporal patterns in our datasets (such as oscillations driven by feedback mechanisms), we clustered log fold-changes from phosphosites that showed significant differences at any time point (adjusted P ≤ 0.05 and |logFC| > 1) compared to unstimulated cells. Silhouette analysis of K-means clustering across different values of K indicated that the best separation was consistently achieved with K = 2 (Supplementary Fig. 2a). This suggests that most EGF-induced changes fall into two main groups over time: up-regulated and down-regulated sites (Supplementary Fig. 2b). For

downstream comparisons across studies, we selected, for each of our studies, the time point with the largest deviation from unstimulated cells, defined as the maximum absolute average log fold change across phosphosites (9 min for HEK293T, 3 min for HEK293F, and 12 min for HEK293F TR).

The studies included in the meta-analysis showed heterogeneity in both the magnitude and significance of phosphoproteomic changes following EGF stimulation (Fig. 2b). In particular, the three DIA HeLa studies (Bortel, Lancaster, and Skowronek) produced broader volcano plots, reflecting stronger overall changes in magnitude compared to the TMT-DDA HEK293 datasets generated in this study, as well as the Tuechler (TMT-DDA) and Chen (SILAC-DDA) datasets. These differences likely reflect the difference in proteomic technologies employed but could also be explained by the underlying biological context, since HeLa and HEK293 cells provide distinct cellular backgrounds. Despite these differences, most phosphosites belonging to the EGFR signaling pathway, as defined in SIGNOR, were consistently up-regulated across all datasets (Fig. 2c). This observation underscores that while experimental platforms and cell types may influence the extent of the

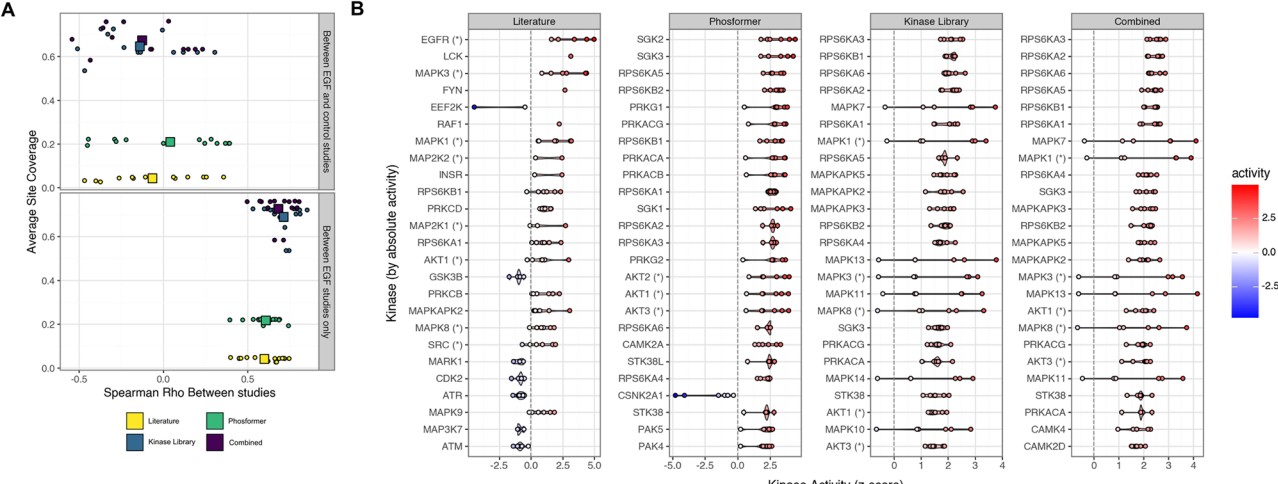

**Fig. 3 | Kinase activity analysis across studies included in the meta-analysis.** **A** Scatter plot showing the relationship between kinase activity correlations and dataset coverage across study pairs. For each point, coverage (Y-axis) represents the proportion of phosphorylation sites covered by a given resource in the first study of the pair, while the X-axis shows the Spearman correlation coefficient (Rho) between the two studies. Facets separate comparisons of EGF studies from those where one study corresponds to a control dataset (Tuechler or Chen). Squares indicate the centroid across the two dimensions per group. **B** Top 25 kinases with the highest absolute average activity per resource for EGF studies. The Y-axis lists the kinases, and the X-axis indicates their estimated activity levels in each study (point). Asterisks indicate kinases that belong to SIGNOR EGFR pathway.

detected changes, the core EGF response remained reproducible across studies.

When examining EGF pathway coverage, we found that the Bortel dataset had the most extensive representation, covering 13 sites, followed by our HEK293 datasets (10–11 sites), the Lancaster study (9 sites), and Skowronek (6 sites). Certain phosphosites exhibited stronger reproducibility than others. For example, Y659 on GAB1, S63 on JUN, and S727 on STAT3 were consistently regulated across nearly all studies, reinforcing their importance as central nodes in the EGF signaling cascade. Others, however, displayed more context-dependent behavior. Notably, T401 on BRAF was down-regulated in HeLa cells but up-regulated in HEK293 cells, pointing to potential cell-line–specific differences in signaling dynamics. Similarly, some EGFR phosphosites, including Y1110 and Y1197, demonstrated robust agreement across datasets, confirming their role as stable markers of pathway activation.

To evaluate agreement between studies beyond EGF pathway sites, we compared all quantified sites using both overlap and correlation measures (Fig. 2d). Jaccard similarity coefficients showed only moderate overlap (0.07–0.43), with the highest value observed between two runs on the same cell line (HEK293F). Spearman correlation scores of fold-changes were also modest (0.01–0.41), and the average correlation between the control studies and the EGF datasets was lower (0.08) than that within the EGF datasets (0.29), indicating greater reproducibility of the EGF signal. When the analysis was restricted to significantly changing phosphosites, reproducibility improved substantially: Spearman's rho values rose to 0.33–0.71, with the strongest agreement seen between the Skowronek and Bortel datasets. No overlapping significantly regulated sites were found for control datasets (Chen and Tuechler) in this analysis. Next, exploiting the time resolution of the HEK293F datasets, we compared log fold-changes of significant sites across time points (Fig. 2e). Here, correlations were higher at early time points (0.64–0.73 up to 15 min) than at the later 25-min time point within the first experiment (X-axis). For the second study, correlation with the first experiment increased with post-stimulation time, from 0.58 at 1 min to 0.7 at 15 min (Y-axis). Together, these results suggest that although overall overlap and agreement are moderate, reproducibility improves when focusing on significantly regulated

phosphosites and is affected by the time of measurement after initial stimulation with EGF.

## Impact of prior-knowledge resources on the identification of EGF-regulated kinases

To explore kinase activity changes, we integrated the meta-analysis data with the kinase–substrate resources and performed kinase activity analysis using differential phosphosite data as input. We also assessed data coverage and compared studies at the kinase activity level (Fig. 3a). Coverage patterns were consistent with Hijazi et al., with the combined resource providing the broadest coverage across studies (70%), followed by the kinase library (64%), Phosformer (21%), and literature (4%). Importantly, for all resources, average correlations of kinase activities were higher between EGF studies than between EGF and control studies (mean Spearman's Rho = 0.65 vs. −0.07, respectively). This indicates that despite variability in coverage, kinase-level agreement across EGF studies remained robust.

We next assessed which individual kinases showed the strongest activity changes across different resources used for the EGF studies (Fig. 3b). The top 25 regulated kinases varied considerably depending on the resource. EGFR, the main receptor kinase activated in response to EGF, was only among the top hits when using the literature-derived resource. By contrast, the Kinase Library primarily highlighted members of the RSK (RPS6KA1–6) and MAPKAPK families (MAPKAPK2/3/5), along with MAPK7, all established downstream effectors of MAPK signaling. Predictions from Phosformer were more diverse, emphasizing SGK family members (SGK1–3), PKA isoforms (PRKACA, PRKACB, PRKACG), an RSK isoform (RPS6KA5), and PKG1 (PRKG1). When combining all resources, the results were again dominated by the RSK family (RPS6KA1–6), underscoring the consistency of these effector modules across methods. In comparison, the literature-based set retained a more established profile centered on EGFR, MAPK1/3, and their classical partners RAF1, INSR, and Src-family kinases (LCK, FYN), as well as RPS6KB1 and PRKCD. Notably, several kinases of the pathway appeared across top hits on multiple resources, including MAPK1, MAPK3, and AKT isoforms (AKT1, AKT2, and AKT3).

Together, these comparisons reveal two complementary pictures of the EGF signaling response. Literature-derived networks recover well-established kinases, consistent with decades of small-scale

studies. In contrast, expanded resources such as Kinase Library and Phosformer point to broader sets of MAPK effector kinases, AGC kinases, and related families that could plausibly play important roles in shaping the downstream signaling landscape. While it remains difficult to validate or reject these expanded views, given that phosphoproteomics data never achieved this level of coverage, it is conceivable that the magnitude of activity changes in these kinases is comparable to those usually associated with EGF response. Thus, the broader resources may be revealing layers of the signaling response that extend beyond the traditional view of the pathway.

### Inference of kinase signaling pathway and ground truth interactions

Up to this point, our analyses have focused on individual sites or kinases. Yet, in many phosphoproteomic applications, a central question is not only which kinases are regulated in response to stimuli, but also how they communicate to trigger a specific biological state (e.g., proliferation or cell death). Such insights reveal pharmacologically relevant targets or uncover alternative signaling routes that cells exploit to develop resistance to treatment. As fully data-driven causal learning would require a vast amount of specific perturbation experiments, currently unavailable for phosphoproteomics, we instead focus on approaches that integrate prior knowledge, in the form of previously validated or predicted interactions, with context-specific kinase activities. We refer to this process as signaling pathway inference (Fig. 4a).

Regarding methods to infer kinase-kinase pathways, we developed three ad-hoc strategies, each of them using the underlying principles of published approaches (Fig. 4b; subnetworks obtained with the three methods using 'literature' as a resource, number of interactions = 10, and data from this study). The first method, inspired by CausalPath[27], scores each interaction by the average change in kinase activity of its source and target ("Source/Target mean activity") and generates a subnetwork from the top-scoring $N$ interactions; this simple baseline uses kinase-level data within a kinase-kinase network but does not incorporate network topology or enforce a directed acyclic graph (DAG). The second method employs network propagation similar to TieDIE[28] and phueGO[29], running a personalized PageRank analysis and selecting top interactions based on the mean PageRank scores of their source and target nodes ("Source/Target Personalized PageRank score", PPR); this approach accounts for network topology and can include kinases without activity estimates if their PageRank scores are high enough, yet it still does not enforce a DAG. The third method applies a "Rooted prize-collecting Steiner tree" (RPCST) formulated as an Integer Linear Programming problem, inspired by TPS[30] and PHONEMES[31], and unlike the others, it guarantees a fully connected subnetwork and enforces a DAG structure, prohibiting feedback loops when given an input root node (here, EGFR). The three methods require three inputs: a kinase-kinase prior network, a list of kinase activities (such as those derived in the previous section), and a specified number of edges. This edge parameter determines how many top interactions are selected in the first two methods and serves as a strict constraint in the RPCST approach, ensuring that all three strategies generate subnetworks of comparable size.

To convert the four kinase-substrate interaction datasets used for kinase activity estimation into kinase-kinase interaction priors, we restricted the interactions to phosphorylation events in which the target site occurs on a kinase protein. Second, to address the large disparity in edge density (the ratio of observed interactions to the theoretical maximum) across resources (1% for the literature-curated network versus 27% for Phosformer, 41% for the kinase library, and 50% for the combined network), we standardized the networks to facilitate fair downstream comparisons. Specifically, we filtered each resource to match the literature network's edge density while preserving

broader kinome coverage. For every kinase in each resource, we selected the top five kinase targets based on interaction weights, which produced networks with edge densities comparable to the literature-based network (Supplementary Fig. 3a). Although these densities were equivalent, the filtered networks encompassed a wider kinome, including 485 kinases in Phosformer, 429 in the kinase library, and 507 in the combined network, compared with 318 in the literature network.

To evaluate the interactions selected by each inference method, we needed to define "ground truth" interaction sets. Establishing this ground truth at the directional interaction level proved considerably more challenging than identifying node-level targets. We constructed three complementary ground truth sets of kinase-kinase interactions. First, we drew on the curated SIGNOR EGFR signaling pathway, which provides well-established, experimentally validated interactions. From this pathway, we extracted only kinase-kinase links, yielding 23 high-confidence interactions (Fig. 4c). Second, we reprocessed the HEK293T "overexpression" dataset from Lun et al.[32], which measured cellular responses to the overexpression of 649 proteins, including all human kinases, across 36 markers before and after EGF stimulation. This perturbational dataset offers a broad but indirect source of interactions where the direction of effect (overexpressed kinase -> measured epitope) is known (Fig. 4d). Third, we generated a context-specific, data-driven set by filtering for functional phosphosites in kinases (using scores from Ochoa et al.[13]) and correlating their abundance across time points in our HEK293F-TR experiment (Supplementary Fig. 3b). To capture possible bidirectionality, we considered each correlated pair in both directions. This third set is referred to as "correlation".

To validate the third strategy introduced in this study, we compared kinase pairs present in the first two ground truth sets with those absent from all sets. Pairs included in the first two sets showed significantly higher absolute phosphosite correlation than the rest (Fig. 4e, one-way ANOVA $p = 0.03$), supporting the assumption that interacting kinases exhibit stronger functional phosphosite correlations. For the second and third resources, we applied three stringency cutoffs: lenient, moderate, and strict (see "Methods" section). When comparing interaction counts and their overlap with the background kinase-kinase networks, the correlation-based set contained the largest number of interactions, followed by the overexpression set, while the SIGNOR set had the fewest and showed no overlap with either the kinase library or the combined resource. Together, these three resources define the "expected" kinase-kinase interactions we aimed to recover through signaling pathway inference.

### Evaluation of inferred signaling pathways

Using estimated kinase activities and kinase-kinase prior networks, we applied three reconstruction methods to build subnetworks of 50-250 interactions, about 5-10% of each full network (1780 edges in the literature set, 1880 in the kinase library, 2331 in Phosformer, and 2457 in the combined resource). These subnetworks capture the most strongly regulated signaling modules by integrating data with prior knowledge. We compared their interactions to the different ground truth sets and recorded the highest overlap across network sizes to account for method differences. This approach ensured that size constraints or alternative optimal solutions did not obscure the best-performing configuration.

Using the SIGNOR pathway as ground truth, the Literature resource consistently recovered more interactions—10-14 across studies and methods—while other resources rarely exceeded four. This enrichment was expected because SIGNOR is partly derived from literature-curated interactions. Differences between EGF-stimulated and control datasets were minimal, with only the Phosformer resource paired with the mean activity method showing a small distinction: control studies recovered one interaction, whereas EGF stimulation yielded up to four. No interactions were recovered when using the

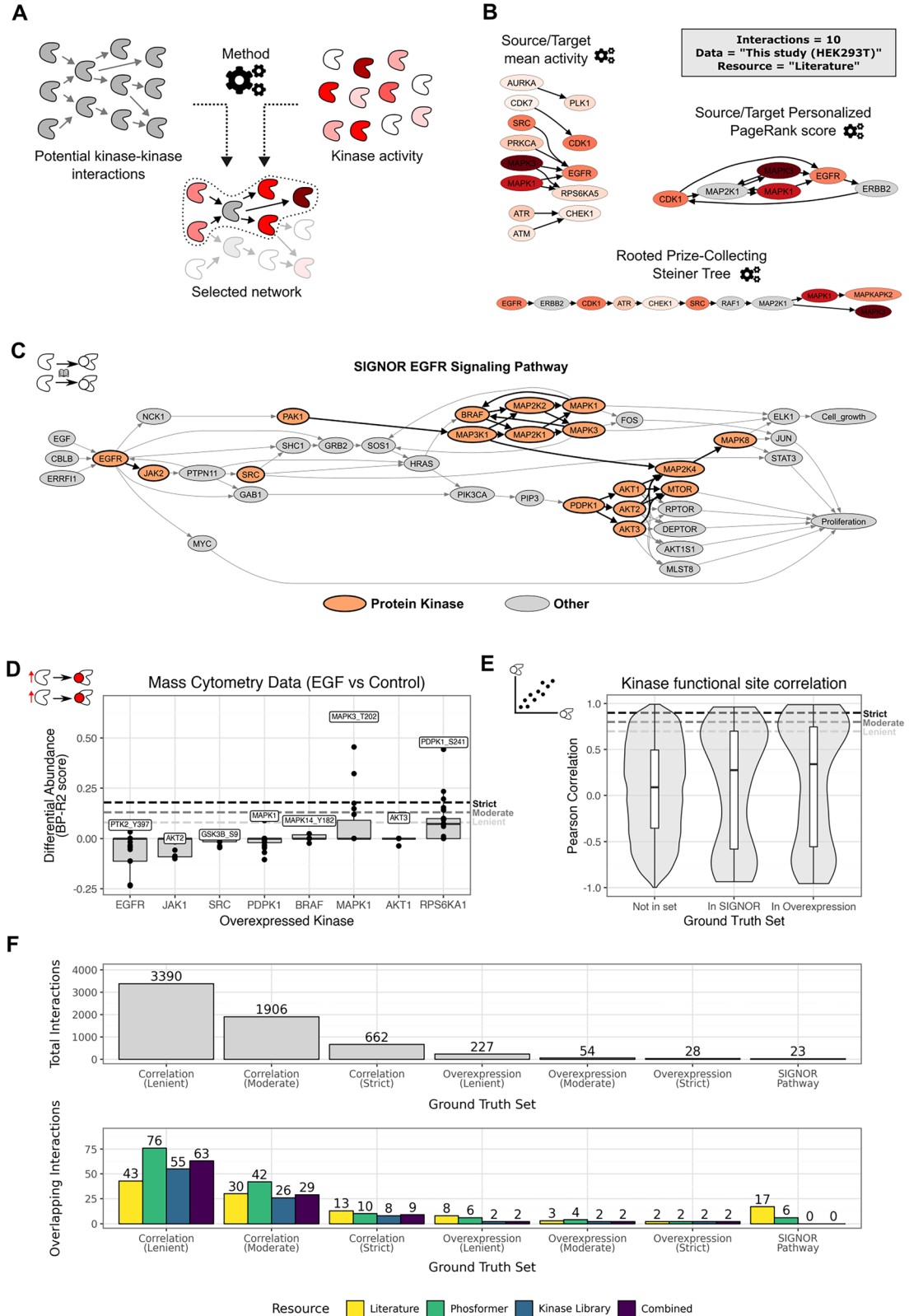

kinase–kinase network from either the kinase library or the combined resource, as expected from the overlaps shown in Fig. 4f.

Literature also dominated the overexpression ground truth (Fig. 5). Only when applying a moderate threshold did Phosformer slightly exceed it, and then by a single interaction. Across methods, PPR recovered more interactions on average (3.6) than mean activity (2.4) or RPCST (2.3). Differences between control and EGF conditions

were again minor, rarely exceeding three interactions. The limited overlap between this ground truth and the kinase–kinase networks likely explains the overall low recovery.

Correlation-based ground truths (lenient, moderate, and strict) reinforced the same trend. Under the lenient threshold, Literature subnetworks recovered up to 29 interactions—more than double any other resource. Phosformer occasionally approached similar

**Fig. 4 | Overview of the signaling pathway inference framework developed in this study and representative results. A** Schematic of the overall concept. **B** Example subnetworks generated by the three inference methods—Source/Target Mean Activity (top left), Source/Target Personalized PageRank Score (right), and Rooted Prize-Collecting Steiner Tree (bottom)—using the literature resource, HEK293T data from this study, and a fixed size of 10 interactions. **C** SIGNOR EGFR signaling pathway from SIGNOR with protein kinases highlighted in orange. **D** Representative differential data from Lun et al., showing BP-R2 scores as boxplots (y-axis) for markers measuring kinase proteins or reporter phosphosites on kinases, with the x-axis indicating the overexpressed kinase in each experiment (N = 38 for all boxplots); dashed lines mark the three cutoffs used here: lenient (0.08),

moderate (0.13), and strict (0.18). **E** Functional phosphosite Pearson correlation values across time points in the HEK293F-TR dataset for kinase pairs not known to interact (N = 86,576), known to interact as per SIGNOR database (N = 112), and which may interact as per the overexpression ground truth set (moderate cutoff) (N = 170); dashed lines indicate the three correlation thresholds: lenient (0.7), moderate (0.8), and strict (0.9). **F** Total number of kinase–kinase interactions per ground truth set (top) and their absolute overlap with kinase–kinase backbone networks from different resources. For the boxplots, the central line depicts the median, the box hinges represent the 25th to 75th percentiles, and the whiskers extend up to 1.5 times the interquartile range above and below the box hinges. Outliers are depicted as individual hollow points beyond the whiskers.

performance in individual studies, while the kinase library and combined networks remained consistently low. EGF stimulation studies produced slightly higher overlaps than controls, but differences remained minimal (average overlap difference = 3.6 in the best settings: Phosformer as resource, Source/Target mean as method, and Correlation (Lenient) as ground truth set). PPR provided a modest but consistent advantage (average overlap = 9.97, compared to 6.70 from Source/Target mean and 5.34 from RPCST), though differences between methods were far smaller than those between resources.

Across all ground truths, the prior network determined recovery far more than the experimental context or reconstruction method. Cell type and stimulation had little effect, and the modest gains from PPR or mean activity compared with RPCST were minor in comparison to the strong advantage of the Literature prior. Overall, the maximum overlap observed was 31 interactions, about 12% of the potential interactions. This highlights that most interactions captured by the subnetworks, data-supported and prior-knowledge coherent, were not present in any ground truth set, illuminating a vast space of kinase-driven signaling interactions that are still to be explored.

## Discussion

In this study, we present data, methods, and evaluation strategies for mapping kinase-to-kinase signaling using phosphoproteomics and kinase–substrate networks. We built comprehensive resources that combine established and predicted kinase–substrate interactions derived from protein language models and peptide-array screens, expanding coverage so that 70–80% of measured phosphosites could be linked to upstream kinases, compared with only about 5% in traditional literature. To examine the EGF pathway, we performed a meta-analysis that merged three published datasets with the three deepest phosphoproteomics experiments generated in this work and added two recent control datasets with different stimuli. From these data, we constructed context-specific signaling pathways using various resource and method combinations and evaluated how well each recovered interactions reported in the SIGNOR EGFR pathway, in experimental overexpression screens, and in correlations of functional phosphosites between kinases. Our analysis showed that resource choice had the greatest influence on the resulting pathways, with Literature outperforming the rest, and that PageRank-based edge selection offers a small advantage compared to the other two methods. Notably, even under the best conditions, only about 90 % of interactions supported by both experimental evidence and prior knowledge were captured in the ground truth sets.

Our work builds on and complements other studies. Invergo et al.[18] focused on predicting signed kinase–kinase regulatory circuits using various data sources, including kinase phosphosite correlation. We built on this concept to establish our third ground truth interaction set. However, their work primarily presents the contextualization of a generic network with data as an application, and they do not explore different methods or compare their results to a context-specific signaling ground truth, which is a key focus of our study. Similarly, Sriraja et al.[19] evaluated signaling network

reconstruction from phosphoproteomics, but used synthetic data for pathway-level analysis and validated data-driven strategies with context-unspecific kinase–substrate interactions. Hill et al.[33] assessed methods for causal reconstruction within the space of dynamic models of cellular signaling, demonstrating that knowledge-assisted approaches generally outperform those that do not utilize prior knowledge, an assumption we adopted in our work. However, their benchmark was confined to just 45 phosphoproteins measured in the study, making it difficult to compare with the broader scale of networks and data used in our work. Hosseini-Gerami et al.[34] evaluated different algorithms for identifying drug mechanisms of action using transcriptomics and prior knowledge as input. Their analysis focused on identifying direct targets and nodes from signaling pathways related to the drug, but transcriptomics, especially after late-stage drug treatment, provides more indirect information compared to short-term phosphoproteomics.

The main challenge in this work was defining a reliable ground truth for kinase–kinase directed interactions. We could partially address it because our study focused on a well-characterized signaling process with a commonly used stimulus and abundant data. Even within the EGF context, however, there is no universally accepted pathway: major knowledge bases such as SIGNOR[35], Reactome[36], and WikiPathways[37] differ in their representations of this signaling cascade. Defining a ground truth would be even more challenging in less-studied stimuli or biological contexts, such as primary cells. In those cases, one option is to pair condition-specific experiments with a broad set of perturbational studies in the same system to identify correlation-supported interactions more reliably, as we did for our third ground truth set. Such data could even support the creation of a context-specific, data-driven prior network to replace the kinase–substrate context-agnostic resources used here. Another strategy is to begin with kinase activity analysis in response to a single perturbation, identify modules of highly deregulated kinases using the workflow presented in this manuscript, and then validate the inferred interactions through targeted, small-scale perturbational experiments, such as kinase inhibition combined with the relevant stimulus.

This study has several limitations. First, while comprehensive, we believe that the number of studies included in the meta-analysis should be increased. In addition to the control studies used here, we expect that the number of publicly available perturbational phosphoproteomics studies will increase thanks to the arrival of protocols such as microPhos[38] and nanoPhos[39]. Second, our network analysis is limited to kinase–kinase interactions, despite the well-established role of other proteins, such as adapters, transporters, and cofactors, in signaling processes. In future developments, we could incorporate these proteins into the kinase-level prior-knowledge networks in combination with technologies like thermal proteome profiling, which would provide functional readouts at scale[40,41]. However, adapting such approaches to the short time scales investigated in this study presents additional experimental challenges.

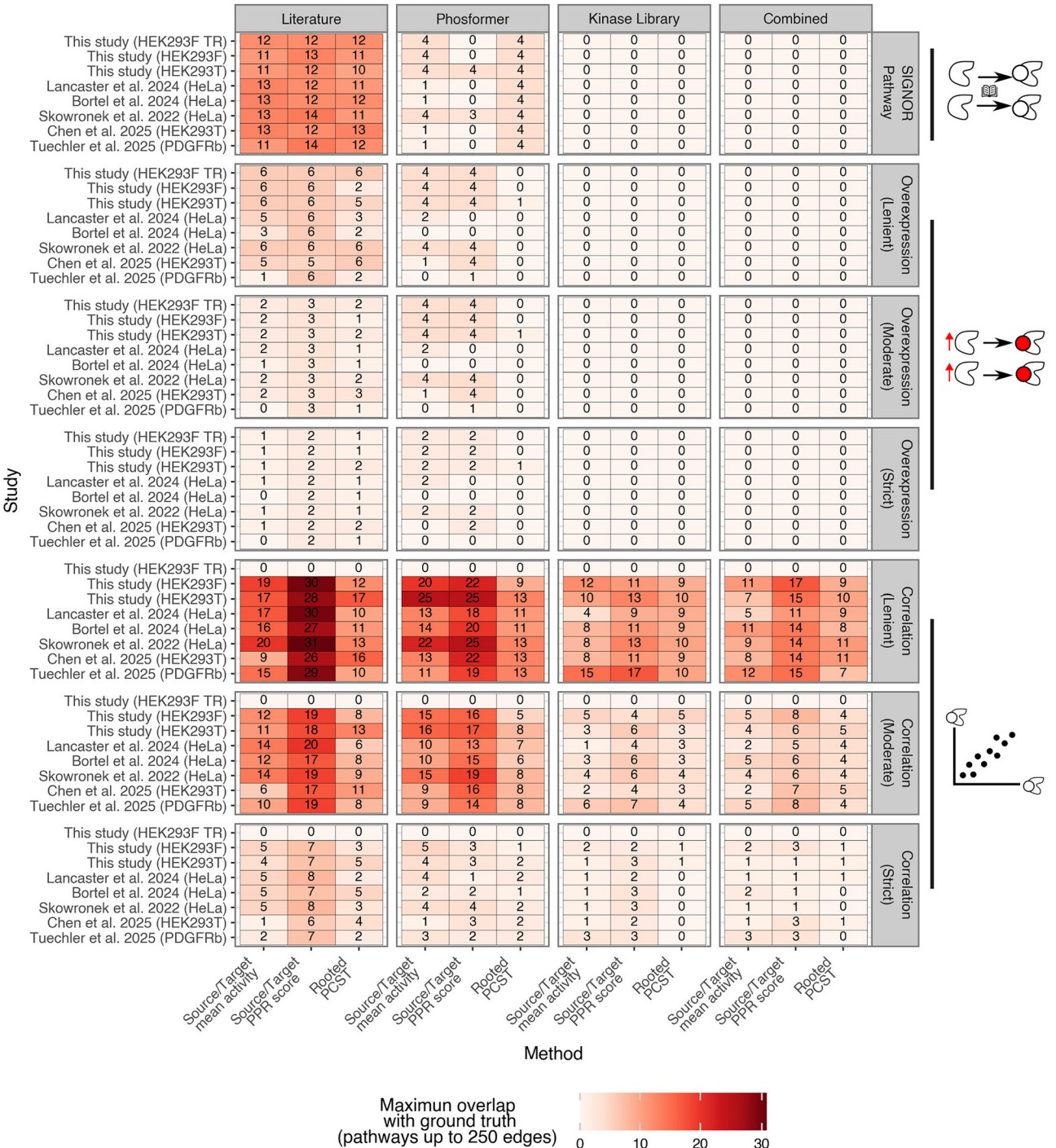

**Fig. 5 | Overlap of reconstructed kinase–kinase subnetworks with ground truth interaction sets.** Heatmaps show the maximum number of overlapping edges between subnetworks (50–250 interactions) inferred from estimated kinase activities and each ground truth network. The x-axis lists reconstruction methods for each prior resource (Mean, PPR, RPCST nested within Literature, Phosformer, Kinase Library, and Combined. The y-axis lists the individual phosphoproteomic studies, including control (Chen and Tuechler) and EGF-stimulated datasets.

Horizontal facets separate the four prior resources, and vertical facets separate the six ground truth sets (SIGNOR pathway; overexpression data at lenient, moderate, and strict thresholds; and correlation-based networks at lenient, moderate, and strict thresholds). Color intensity indicates the number of recovered interactions. When the correlation ground truth was used, overlap with 'This study (HEK293F TR)' was set to zero, as the dataset was used to create the ground truth.

In summary, this study expands the traditional understanding of signaling pathways by leveraging state-of-the-art resources, data, and computational methods. Only a small proportion of the interactions selected by the methods overlapped with the different ground truth sets, suggesting that we need to start considering the broader signaling possibilities offered by modern phosphoproteomics and computational methods. Ultimately, this expanded knowledge should lead to the development of more effective molecular interventions, especially in contexts where EGF signaling plays a fundamental role, such as cancer. Specifically, we envision this knowledge as the basis for the rational design of more effective anti-cancer combination therapies[42].

## Method

### Cell culture, treatment, and cell lysis

All cell lines used in this study were verified to be negative for Mycoplasma contamination. HEK293T cells (obtained from ATCC, catalog number: CRL-3216) were cultured in DMEM (Sigma-Aldrich, D5648) containing 4.5 mg ml$^{-1}$ glucose, 10% (vol/vol) FBS (Gibco, 10270), and 1 mM L-glutamine (Gibco, 25030081) at 37 °C with 5% $CO_2$. Expi293F (Thermo Fisher Scientific, catalog number: A14527), also referred to as HEK293F throughout the manuscript, were cultured in FreeStyle™ 293 Expression Medium (serum-free medium), at 37 °C with 5% $CO_2$, shaking at 125 rpm. For Expi293F cells, 5 ml aliquots (with $1 \times 10^6$ cells/ml) were treated with 0.1 μg/μl EGF (biotechne) for the indicated times. Stimulation was stopped by adding 45 ml of ice-cold PBS and placing the samples on ice. Cells were spun down at $300 \times g$ for 5 min, followed by flash freezing of the cell pellet in liquid nitrogen before sample lysis. For HEK293T cells, 0.5 million cells were seeded in 150-mm dishes and grown for 3 d. One hour before the EGF treatment, the culturing medium was replaced with 15 mL DMEM (Sigma-Aldrich, D5648) containing 4.5 mg ml$^{-1}$ glucose, 2% (vol/vol) FBS (Gibco, 10270), and 1 mM L-glutamine (Gibco, 25030081). Per replicate and condition, one dish was treated with 0.1 μg/μl EGF (biotechne) for the indicated times. Stimulation was stopped by placing the dishes on ice, removing the culturing medium, and washing cells with 15 mL of ice-cold PBS per dish twice. Cells were harvested by scraping, and pelleted at 4 °C at $300 \times g$ for 5 min before lysis. The experiments included the following replicates, time points, and TMT sets: HEK293F – $N = 4$ with three time points (3, 9, and 25 min, TMT16); HEK293F – $N = 4$ with three time points (TMT16); and HEK293F-TR – $N = 2$ with nine time points (TMT-18).

The lysis buffer consisted of 4 M guanidinium isothiocyanate, 50 mM HEPES (2-[4-(2-hydroxyethyl)piperazin-1-yl]ethanesulfonic acid), 10 mM TCEP (tris(2-carboxyethyl)phosphine), 1% N-lauroylsarcosine, 5% isoamyl alcohol, and 40% acetonitrile, with the pH adjusted to 8.5 using 10 M NaOH. For sample lysis, a volume of buffer equivalent to approximately 5 times the volume of the cell pellets was used. The samples were homogenized by pipetting, incubated at room temperature on a shaker for 15 min and centrifuged at $16,000 \times g$ for 10 min at room temperature to remove cell debris and nucleic acid aggregates. Protein concentrations were measured using a tryptophan fluorescence assay, following the method of Wisniewski et al.[43]. Samples were transferred to multiscreenHTS-HV 0.45 μm 96-well filter plates with PVDF membranes (Merck Millipore), and ice-cold acetonitrile was then added to the samples to induce protein precipitation, reaching a final concentration of 80% acetonitrile. After 10 min of incubation, the samples were centrifuged, and the solution was removed. The protein precipitates were washed twice with 200 μL of 80% acetonitrile and twice with 200 μL of 70% ethanol, centrifuging at $1000 \times g$ for 2 min for each wash.

Next, a digestion buffer containing 100 mM HEPES (pH 8.5), 5 mM TCEP, 20 mM chloroacetamide, and trypsin (TPCK-treated, Thermo Fisher Scientific) was added to the protein precipitates. The trypsin-to-protein ratio was set to 1:25 (w/w), with a maximum final protein concentration of 10 μg/μL. Tryptic digestion was performed overnight at room temperature with mild shaking (600 rpm). After digestion, the samples were spun down, acidified to 1% TFA, and desalted using Sep-Pak tC18 columns (Waters), eluted with 0.1% TFA in 40% acetonitrile, and dried using a vacuum concentrator before phosphopeptides enrichment.

### Phosphopeptide enrichment

Lyophilized peptides were resuspended in loading and washing buffer (80% acetonitrile, 0.07% TFA), sonicated, and centrifuged at 16,000 x g. Phosphopeptide enrichment was performed as described in Leutert et al.[44] using the KingFisher Apex robot (Thermo Fisher Scientific) with 50 μL of Fe-NTA MagBeads (PureCube) per sample. After five washes with buffer A, bound phosphopeptides were eluted by adding 100 μL of 0.2% diethylamine in 50% acetonitrile, followed by lyophilization.

For TMT labeling, the enriched phosphopeptides were resuspended in 10 μL of 100 mM HEPES (pH 8.5), and 4 μL of TMTPro reagent (20 μg/μL in acetonitrile) was added. The labeling reaction proceeded for 1 h at room temperature, after which it was quenched by the addition of 5 μL of 5% hydroxylamine for 15 min. Labeled peptides from the same experiment were pooled and lyophilized. Before fractionation, the labeled phosphopeptides were resuspended in 50 μL of 10% TFA and desalted using in-house C18 stage tips[45] packed with 1 mg of ReproSil-Pur 120 C18-AQ 5 μm material (Dr. Maisch) above a C18 resin plug (AttractSPE disks bio - C18, Affinisep).

### PGC-LC offline fractionation

The samples were reconstituted in 18 μL of buffer A (0.05% TFA in MS-grade water with 2% acetonitrile), with an injection volume set to 16 μL. Phosphopeptide separation was performed using a Hypercarb column (100 mm length, 1.0 mm inner diameter, 3 μm particle size, Thermo Fisher Scientific) at 50 °C, with a flow rate of 75 μL/min, on an Ultimate 3000 Liquid Chromatography system (Thermo Fisher Scientific). The linear gradient separation began 1 min post-injection, increasing from 13% buffer B (0.05% TFA in acetonitrile) to 42% buffer B over 95 min, followed by an increase to 80% buffer B within 5 min. The column was washed with 80% buffer B for 5 min and then re-equilibrated with 100% buffer A for 5 min. Fractions were collected from 4.5 to 100.5 min at 2-min intervals, producing 48 fractions, which were then pooled into 24 by combining each fraction with its corresponding $n + 24$ fraction. The samples were dried using a vacuum concentrator prior to LC–MS/MS analysis. For the HEK293F-TR dataset, samples were separated into 96 fractions, collected every 2 min (with a linear gradient from 13% to 42% buffer B), and then pooled into 48 fractions.

### LC−MS/MS analysis

All samples were resuspended in a loading buffer containing 1% TFA, 50 mM citric acid, and 2% acetonitrile in MS-grade water. Liquid chromatography separation was carried out using an UltiMate 3000 RSLCnano system (Thermo Fisher Scientific). Peptides were first trapped on a cartridge (Precolumn: C18 PepMap 100, 5 μm, 300 μm i.d. × 5 mm, 100 Å) before separation on an analytical column (Waters nanoEase HSS C18 T3, 75 μm × 25 cm, 1.8 μm, 100 Å). Solvent A consisted of 0.1% formic acid with 3% DMSO in LC–MS-grade water, while solvent B contained 0.1% formic acid with 3% DMSO in LC–MS-grade acetonitrile. Peptides were loaded onto the trapping cartridge at 30 μL/min with solvent A for 5 min, then eluted at a constant flow rate of 300 nL/min. The peptides were separated using a linear gradient of buffer B, from 7% to 27%. This was followed by an increase to 40% buffer B within 4 min, a wash at 80% buffer B for 4 min, and re-equilibration to initial conditions.

The LC system was coupled to either a Fusion Lumos Tribrid or an Exploris 480 mass spectrometer (Thermo Fisher Scientific), operated in positive ion mode with a spray voltage of 2.4 kV and a capillary temperature of 275 °C. Full-scan MS spectra were acquired in profile mode using the Orbitrap with a resolution of 90,000 and a mass range of 375–1500 m/z. The maximum injection time was set to 50 ms, and automatic gain control (AGC) was set to $4 \times 10^5$ charges (Fusion Lumos) or $3 \times 10^6$ charges (Exploris 480). The mass spectrometers were operated in data-dependent acquisition mode with a maximum duty cycle time of 3 s, selecting precursors with charge states 2–7 and a minimum intensity of $2 \times 10^5$ for subsequent HCD fragmentation. Peptide isolation was performed using the quadrupole with a 0.7 m/z isolation window. Precursors were fragmented using a normalized collision energy of 32% or a stepped collision energy of 31 ± 3%. A dynamic exclusion window of 30 s was applied, and MS/MS spectra were acquired in profile mode at a resolution of 50,000 (Fusion Lumos) or 45,000 (Exploris 480) using the Orbitrap,

with a maximum injection time of 100 ms and an AGC target of $1 \times 10^5$ charges.

## LC−MS/MS data analysis

Raw files were converted to mzmL files using MSConvert from Proteowizard[46], using peak picking and keeping the 1000 most intense peaks per spectrum. Files were then searched using MSFragger v4.0[47] in Fragpipe v21.0 against the SwissProt Homo sapiens database (20,443 entries). The default TMT16-phospho workflow was used, with a few modifications: oxidation on methionine (maximum 2 occurrences), protein N-terminal acetylation (maximum 1 occurrence), phosphorylation on S/T/Y (maximum 3 occurrences), and peptide N-terminal TMT16 labeling (maximum 1 occurrence) were set as variable modifications, with a total of up to 5 variable modifications allowed per peptide. Lysine TMT16 labeling and cysteine carbamidomethylation were set as fixed modifications. Percolator was used for PSM validation, and PTMProphet was used to determine site localization. A FDR cutoff of 1% was used. For the TMT quantification, Philosopher was used to extract MS1 and TMT intensities.

The peptide spectrum match (PSM) tables produced by FragPipe were filtered to keep only PSMs mapping to unique phosphopeptides having a purity value ≥ 0.5. Then, we summarized information at the phosphosite level (only sites with a phosphorylation localization probability ≥0.75 were kept for subsequent analysis) by summing the TMT intensities of all the PSMs assigned to a specific phosphosite.

## Differential-abundance analysis

Site-level PSM tables were then used as input for differential-abundance analysis. Intensity tables were normalized using variance stabilization normalization via the vsn R package (v3.70.0)[48]. Then, normalized intensities were compared between treated and untreated samples using the limma R package (v3.58.00)[49]. Samples from each time point were compared against the untreated samples, resulting in differential-abundance estimates for each time point. For the HEK293T data, one outlier sample per time point was removed before the differential-abundance analysis, resulting in three samples per time point. All *P*-values from the differential-abundance analysis were adjusted for multiple comparisons using the Benjamini and Hochberg method[50].

## Kinase−substrate networks

The list of human kinases was obtained from the CORAL GitHub repository [https://github.com/dphansti/CORAL/blob/master/Data/kinmaplabels.txt]. CORAL was also used to visualize targets across the human kinome[51]. Literature-based interactions were sourced from the OmniPath Enzyme-Substrate dataset (snapshot from 18-09-2024), including all phosphorylation-related interactions from PhosphoSitePlus[52] and SIGNOR[35]. Specificity scores were gathered from the Serine/Threonine kinase library (Supplementary Table 3 in Johnson et al.[15]) and the Tyrosine kinase library (Supplementary Table 3 in Yaron-Barir et al.[16]).

For computational predictions, we used Phosformer[14], downloading and running the model via the Python module [https://github.com/esbgkannan/phosformer]. All human phosphosites listed in PhosphoSitePlus (240,165 sites, snapshot from 01-09-2024) and their corresponding 11-mer sequences, centered on the phosphorylation site with six flanking amino acids on either side, were used as potential targets.

Three cutoffs were investigated for both the Kinase Library and Phosformer interactions:

Lenient: ≥0.5 for Phosformer and ≥90 for Kinase Library

Moderate: ≥0.65 for Phosformer and ≥95 for Kinase Library
Strict: ≥0.8 for Phosformer and ≥99 for Kinase Library

For each cutoff level, a combined network was also generated by taking the union of predictions from the literature, Phosformer, and the Kinase Library. For each cutoff on the Kinase Library and Phosformer networks, scores were rescaled to the 0−1 interval by dividing by the maximum probability or specificity score.

Hijazi et al.[21] data to evaluate the signal on different kinase−substrate networks was retrieved from the GitHub repository accompanying the manuscript: [https://github.com/CutillasLab/ebdt/tree/master/requiredData]. For each drug, we used kinases inhibited >50% in the kinaseInhibitionSpecificity.csv dataset as true positives and all other screened kinases as true negatives. To test the effect of different thresholds, we also evaluated cutoffs ranging from 10% to 80% in 10% increments. Next, kinase activity analyses were performed using site-level log fold-changes as input, along with kinase−substrate networks, applying the Z-score method implemented in the decoupleR Python module (v1.8.0)[53]. Kinases with at least five measured target phosphorylation sites were considered. Area under the receiver operating characteristic curve (AUROC) scores were calculated as described by Müller-Dott et al.[22]. Briefly, for each network, kinase activities across experiments were combined. In contrast to Müller-Dott et al., only kinases with an estimated activity score in all resources were used in this analysis. 1000 random subsamples were generated to balance true positives and true negatives. Negative kinase activities were then used as predictive vectors to obtain AUROC scores.

To assess sequence similarity between kinases, we used MMSeqs (v18.8cc5c)[54]. Two kinases were classified as "similar" if their sequence identity was at least 50%. For the specialized versus convergent phosphosite analysis, we applied a cutoff of three kinases: sites with more than three upstream kinases were defined as convergent, whereas sites with fewer than three were defined as specialized. Overlap coefficients were calculated between all pairs of kinases within a given resource by dividing the number of shared target phosphosites by the minimum number of targets in either kinase target set.

The kinase−substrate networks described in this manuscript are available through the accompanying Python package. Use kinsub.get_combined_kinsub() to retrieve networks across all cutoffs, kinsub.get_one_cutoff_combined_kinsub() to obtain the network for a specific cutoff defined above.

## Meta-analysis of EGF phosphoproteomic studies

The selection of recent EGF studies for the meta-analysis was based on the following criteria: (1) The study had been published in a peer-reviewed journal, (2) Raw data was publicly available through a proteomic repository such as PRIDE, and (3) The study had been published within the last three years.

Data from Skowronek et al.[11] were downloaded from the PRIDE repository ([https://ftp.pride.ebi.ac.uk/pride/data/archive/2022/08/PXD034128/Phospho_biological_study_post-analysis_files.zip]). Peptide intensities were summed at the site level and filtered to retain only sites with information in at least 2 untreated and 2 EGF-treated samples. Differential expression analysis was then performed using VSN normalization and limma, as described above.

Data from Bortel et al.[10] for the larger tested sample size (12 samples) were provided by the authors. Peptides were filtered to those with site localization probability >75, and the intensities of all peptides covering a specific site were summed. Sites with information in at least 2 untreated and 2 EGF-treated samples were retained. Differential expression analysis was then conducted using VSN normalization and limma.

Data from Lancaster et al.[24] for the Astral runs were retrieved from Supplementary Table 2 of the paper. Peptides with data in at least 2 replicates per condition were filtered, and the intensities were summed at the site level. Differential expression analysis was subsequently carried out using VSN normalization and limma.

Data from Tucheler et al.[26] were obtained from the manuscript's GitHub repository: [https://github.com/saezlab/kidneyfibrosis_multiomicsmodel_paper]. Specifically, we downloaded the differential-abundance analysis Rds file and used it for subsequent analyses (2024-08-15_diff_results.RData).

Data from Chen et al.[25] were obtained from Supplementary Table 4 (mmc10.xlsx) of their manuscript. All downstream analyses used the data contained in the Combined test result tab.

To compare identified sites between studies, we used the Jaccard Similarity index, which divides the size of the intersection of two sets (in this case, phosphosites identified in each study) by the size of their union.

All kinase activity analyses were performed using site-level log fold-changes as input, along with kinase–substrate networks, applying the Z-score method implemented in the decoupleR Python module (run_zscore, v1.8.0)[53]. Kinases with at least five measured target phosphorylation sites were considered.

A master table of site-level differential-abundance changes, covering both the data presented in this manuscript and all studies included in the meta-analysis, can be accessed through the accompanying Python package. Use the function phosphodata.get_all_studies() to retrieve it.

### Context-specific kinase signaling pathway inference

To infer kinase-driven signaling pathways, kinase–substrate networks were converted into kinase–kinase networks. This was done by filtering the interactions to retain only those where both the source and target were kinases, as defined by CORAL annotations (see above). Interaction weights were assigned based on different criteria: the number of references for the literature network, the maximum probability for Phosformer, the maximum specificity for the kinase library, and a combination of these for the combined network.

To reduce the density of the Phosformer, kinase library, and combined networks, filtering methods were tested to retain the top target kinases for each source kinase based on interaction weight. Retaining between 3 and 250 top targets was evaluated, and the top 5 were ultimately selected, as this most closely matched the edge density of the literature network.

The context-specific signaling pathway inference was performed for every possible combination of contrast (which can include multiple contrasts within a study) and kinase–substrate networks (resources). For each contrast, inferred kinase activities were converted to absolute values and normalized using the maximum absolute activity, resulting in values ranging from 0 to 1. These scores were then used as input data for the various methods outlined below.

**Mean**. All kinase–kinase interactions for a given resource were scored by the mean normalized absolute activity of the source and target kinases. The top $N$ interactions were then selected and used to construct a subnetwork. Only interactions between kinases with an estimated kinase activity were considered in this method. In the weighted version, the average kinase activities were multiplied by the edge weights, which ranged from 0 to 1.

**PageRank**. For propagation-based network contextualization, personalized PageRank (PPR), as implemented in the NetworkX Python module (v3.2.1), was applied to the kinase–kinase networks. The personalization vector, which determines the probability of the random walker revisiting specific nodes, was provided using the normalized kinase activity vectors of the kinases. A default damping factor of 0.85 was used for the first analysis, and a range from 0.5 to 0.99 was used in the second experiment. The resulting PageRank scores were then used instead of the kinase activity values to retrieve the stop $N$ interactions with higher average PageRank values. In contrast to the mean method, kinases for which kinase activity could not be inferred were also included. In the weighted version, the PPR algorithm was provided with the weights during the calculation of the scores.

In the randomized input experiment, the labels of the personalization vector were shuffled before running PageRank and selecting interactions. This process was repeated 1000 times. In the missing data experiment, a probability vector, derived from the min-max normalized absolute kinase activities, was used to generate subsamples of the kinase activity vectors. These subsamples contained varying proportions of the original kinases, ranging from 10% to 90%, tested at 10% intervals.

**Rooted Fixed-Edge Prize-Collecting Steiner Tree (RPCST)**. For the rooted fixed-edge prize-collecting Steiner tree, the Integer Linear Programming (ILP) constraints defined in CORNETO[55] were reused and extended to meet the specific needs of this study. Briefly, the ILP constraints ensure that: (1) If an edge is selected, both the source and target nodes are selected, (2) If a node is selected but is not the root, it must have at least one incoming edge, (3) All nodes included that are not in the input measurements must have at least one outgoing edge if selected. Here, the input measurements are the normalized kinase activities. (4) No loops are present in the solution, and (5) A fixed number of edges is selected. The objective function for the optimization is to recover the maximum amount of signal, defined as the sum of the absolute kinase activities. In the weighted version, inverse edge weights were also added as a second term to the objective function so that the tree with the lowest edge weight is selected. The ILP problem was formulated using the CVXPY Python module (v1.5.0). The mathematical optimization toolkit GUROBI (v11.0) was then employed to solve the problems using a time limit of 30 s and mipGap = 0.05. EGFR was used as the root node.

All network-inference methods are provided in the network_methods module of the phosphonetworks package. The three functions: mean_selection(), pagerank_selection(), and rpcst_selection(), each accept a kinase–kinase template network as a networkx object, a dictionary of terminals formatted as (Kinase: Value), and a specified number of edges, and return the selected network.

### EGF ground truth interactions

Pathway kinase–kinase interactions were obtained from SIGNOR ([https://signor.uniroma2.it/pathway_browser.php?pathway_list= SIGNOR-EGF]). Kinase overexpression mass-cytometry BP-R2 scores from Lun et al.[32] were downloaded from Supplementary Table 4 of the publication. Differential analysis of EGF-treated versus untreated samples was performed following the methodology described in the original publication. Briefly, BP-R2 scores were filtered to retain those higher than the maximum score found for controls (untransfected or transfected with FLAG-GFP). The difference in BP-R2 scores between EGF-treated and untreated samples was then assessed, with values greater than 0.08 (Lenient), 0.13 (Moderate), and 0.18 (Strict) being considered as significant hits in response to EGF treatment for a given kinase overexpression experiment.

To generate correlation-supported interactions, HEK293F-TR data was filtered to retain only functional sites in kinases (defined as those with a functional score ≥0.5 from Ochoa et al.[13]). Next, we correlated their differential abundance across time points and retained interactions between kinases supported by functional phosphosites with a Pearson correlation score ≥0.7 (Lenient), 0.8 (Moderate), and 0.9 (Strict).

### Reporting summary

Further information on research design is available in the Nature Portfolio Reporting Summary linked to this article.

## Data availability

All the data to reproduce the results presented in this study can be retrieved from Zenodo: [https://zenodo.org/records/18390833]. The mass spectrometry proteomics raw data generated in this study have been deposited to the ProteomeXchange Consortium via the PRIDE[56] partner repository with the dataset identifier PXD056666.

## Code availability

All the code to reproduce the results of this manuscript can be accessed at: [https://zenodo.org/records/18390833].

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

## Acknowledgements

MGR was supported through state funds approved by the State Parliament of Baden-Württemberg for the Innovation Campus Health + Life Science Alliance Heidelberg Mannheim. We thank Ana Mellado-Fuentes for her assistance with the HEK293T experiment and Attila Gabor for fruitful discussions. We thank the authors of studies re-analyzed in this manuscript for providing transparent access to their data. We acknowledge EMBL IT Services for support with high-performance computing.

## Author contributions

M.G.R.: Conceptualization, Data Curation, Formal Analysis, Investigation, Methodology, Project Administration, Software, Writing - Original Draft Preparation. C.P.: Conceptualization, Investigation, Methodology, Validation, Writing - Original Draft Preparation. M.L.B.: Investigation, Visualization. I.B.: Investigation, Visualization. P.R.M.: Methodology, Software, Formal analysis. S.M.D.: Software, Formal analysis. M.S.: Resources, Project administration, Writing - Original Draft Preparation, Supervision, Funding acquisition. J.S.R.: Resources, Project administration, Writing - Original Draft Preparation, Supervision, Funding acquisition.

## Funding

## Competing interests

J.S.R. reports funding from GSK, Pfizer, and Sanofi and fees/honoraria from Travere Therapeutics, Stadapharm, Astex, Pfizer, Grunenthal, Moderna, and Owkin. The remaining authors declare no competing interests.
