## [Transparent Peer Review file · Nature Communications]

Benchmarking EGF signaling pathway inference using phosphoproteomics and kinase-substrate interactions

Corresponding Author: Professor Julio Saez-Rodriguez

Version 0:

Reviewer comments:

Reviewer #1

(Remarks to the Author)

Garrido-Rodriguez et al., focuses on context-specific signaling pathway inference from phosphoproteomics datasets (online and in-house) and kinase-substrate networks. They revised EGFR signalling and proposed that literature-curated networks combined with network propagation yield the best recovery of signalling interactions. Finally, they claim that this analysis will help finding novel treatments against EGFR-driven diseases.

The manuscript is a novel addition to a series of emerging studies (listed and well referenced by the authors) that aim to challenge the status quo of the “traditional understanding of signaling pathways”. It expands on several fronts, most importantly, by 1) analysing short-term stimulated phosphoproteomes, 2) testing several computational methods, and 3) comparing the results to multiple well-defined ground truth sets.

However, a few concepts throughout the manuscript need clarification or more evidence especially for non-expert readers. The method described here have a great potential to be used by signalling experts and computational biologists but requires more qualitative examples addressed to a broad readership.

The idea of combining available information on kinase-substrate relationship and in-house generated datasets is modern and well executed. However, I have the following comments:

Major comments:

1. Results. From “To determine” on page 5 until “compared to literature” on page 6 there are only two panels (Fig 1A and Fig S1A). A workflow is needed as well as some more illustrations on what the author did to come to this first conclusion.
2. Results. “far exceeded the known kinase-substrate interactions from the literature”. This is not shown anywhere I can see. Please, provide the evidence for this sentence.
3. Results: “kinases with similar evolutionary background do not share a high amount of targets”. Provide references and/or evidence.
4. Results: “As expected, adding new interactions reduced the frequency of specific targets, defined as phosphosites regulated by a single upstream kinase”. Again, this is not clear and not supported by evidence.
5. Results, beginning of page 7: “uncovering new interactions across various kinase families.”. There are neither quantitative nor qualitative evidence.
6. Results. Where do you show “state-of-the-art kinase-substrate interactome”?
7. Results: how do they define canonical EGFR pathways? Please, clarify
8. Results: after presenting the workflow in Figure 4 the authors only describe the different steps of their analysis and the comparison between datasets and type of data. What is missing are examples at each step or, in other terms, a qualitative evaluation of their main conclusions. Which kinase-substrate are they talking about? This would enormously help the reader (and ultimately, the users of their methods) to apply these methods to other datasets
9. Figure 2D-E. The results should be discussed in more detail, as the large differences/variance among different studies is critical for the interpretation of later results.
10. Figure S3C. The difference between the 3 different network inference algorithms would be easier to understand if

Supplementary Figure S3C was included in the main text along with the explanation of differences in topology. This would also make this part more accessible to a more general audience.

11. Discussion: how would the authors envision to apply their methods to signalling pathways less known or understudied compared to the EGFR pathway and in more relevant cell models or primary cells? They present limitations of their study, but these points require more discussion

12. General comment: Why did the authors choose to use different time points of EGF stimulation? It is well-known that (among others) the duration of stimulation drives different cellular responses. This makes the comparison of these time-resolved datasets to literature datasets with a single time point problematic, which is not addressed in the manuscript.

13. General comment. For the generalizability of the results, the inclusion of only two cell lines is not enough, however, it makes the results harder to interpret. Did the authors test whether only including data from HEK293 cells (data from this study) or data from HeLa cells (literature data) changes the results/optimal method?

Minor comments

1. Introduction. The sentence "Such an analysis would allow us to quantify how our traditional view of signaling pathways aligns with the broader patterns of activation revealed by new methods with larger coverage" require clarification and contextualization.

2. Introduction. EGFR signalling is crucial in breast, lung, melanoma and at least also glioblastoma.

3. Results: in the sentence, "This approach allowed us to generalize known interactions from literature to kinases and substrates with similar evolutionary backgrounds." It is unclear what "similar evolutionary background" refers to.

4. Results: the three Excel files submitted as supp material are not cited in the text. It is not clear what is shown in each of them

5. The results on Figure S2C should be included in the main text and discussed

(Remarks on code availability)

Reviewer #3

(Remarks to the Author)

In this manuscript, Garrido-Rodriguez, Potel and colleagues explore different ways to infer pathways from large-scale phosphoproteomics data, using both new phosphoproteomic data and existing datasets. They compare different pathway inference methods and different ground truth sets to evaluate the performance of pathway inference, focusing on the well-characterized EGF/EGFR pathway. The topic is certainly important, since we are being inundated with large-scale datasets from different sources, and making biological sense of these datasets is of utmost importance. At the same time, it is a big service to the community to evaluate different methods and suggest gold standard approaches for analyzing large-scale (phosphoproteomics) data.

However, after reading this manuscript as a biologist who is interested in signaling networks and interactions, I am not sure how much I have learnt. To be clear, I cannot comment on the methodological aspects or the computational analysis (although I have some general questions/suggestions below), but as I was trying to distill the results of the manuscript to an actionable output or to a fundamentally novel insight, I had hard time doing so. Perhaps this is due to the writing style that is not very accessible to an experimentalist and lists results in a matter-of-fact manner without highlighting the most relevant insights. Overall, I am not quite sure if this manuscript moves the needle enough to warrant publication in its current form in Nature Communications.

General comments

1. It would be useful to include full ROC curves in the supplement, as the differences between the different resources are relatively modest by just looking at AUROC (~0.60 to ~0.68 in Fig 1A)

2. It is intriguing that Phosformer is so highly enriched for tyrosine kinases (Fig 1E), which are arguably the best-characterized kinases due to their importance in tumorigenesis and the availability of many specific inhibitors. I wonder how much this difference contributes to differential performance of different kinase-substrate networks. For example, does Phosformer perform better (or worse) if the analysis is limited to tyrosine kinases?

3. The authors use three previously published EGF phosphoproteomics datasets in HeLa cells and generate three other datasets in HEK293 cells. It is not clear why HEK293 cells were selected as a model (they express very low amounts of EGFR) in contrast to other cell lines that have much higher EGFR expression.

4. It would be useful to add another phosphoproteomic dataset or datasets as a control for specificity. For example, what would the Jaccard similarity look like if similar datasets for other RTKs or e.g. TGFbeta signaling were included as outgroups/controls?

5. Did the Jaccard similarity calculation only consider phosphopeptides that were detected in both studies or did it consider all phosphopeptides? The former would be a better measure of the similarity of the biological responses, since mass spec instruments and experimental conditions affect the sensitivity of each assay differently.

6. What is absolute difference in kinase activity (page 12 and Fig 3C)? This is poorly described in the results section and therefore interpretation of the results is quite difficult.

7. The authors mention that in the HEK293F-TR experiment, the peak of activation (absolute difference in kinase activity?) shows two peaks, which was absent in non-time resolved studies. Looking at the data in Fig S2C, the activity in HEK293F (middle panel) seems to show an opposite trend compared to HEK293F-TR (left panel). In the HEK293F experiment, 3-min timepoint shows lower activity than the 9-min timepoint, whereas in the HEK293F-TR experiment, the activity is higher at 3 minutes and lower in 9 minutes. If the 3-min and 9-min experiments show such a different pattern although they should (to my understanding) be essentially identically performed, how confident can one be about these patterns?

8. The kinase activity analysis in Fig 3D: I'm a bit puzzled about the results and interpretation. If the cells are stimulated with EGF, shouldn't EGFR be the top kinase in any analysis that is biologically relevant? The fact that only one method (literature) actually captures what is happening at the cells is alarming. Although it is certainly true that many other kinases are activated downstream of EGFR, it is biologically unlikely that they would be activated more in terms of fold-changes in target phosphorylation and/or in terms of total fraction of targets regulated. As an end user, one would hope that in this extremely well controlled situation where only a single ligand is added, the primary target of the signaling cascade would light up in the analysis. If this is not the case, what hope do we have for all the murky biology that happens in more natural contexts? I would really appreciate if the authors commented and discussed this.

9. Finally, it would be useful to know how sensitive the authors' analysis is to the selected cutoffs (which are explained in the methods section) in the used datasets such as Hijazi et al. and the ground truth sets (e.g. kinase overexpression).

Minor comments

10. The authors often refer to "deregulation" of signaling when analyzing EGF-induced changes. Although this is semantics, isn't this just "regulation" since there is nothing abnormal about EGF inducing phosphorylation changes via EGFR?

11. Fig S4B (page 16) should be Fig S5B

(Remarks on code availability)

Version 1:

Reviewer comments:

Reviewer #1

(Remarks to the Author)

The authors have answered to my questions and clarified all the requested points.

(Remarks on code availability)

Reviewer #3

(Remarks to the Author)

The authors have addressed all my comments in the revised version by conducting further analyses and adding control datasets. I have no further comments!

(Remarks on code availability)

Reviewer #1 (Remarks to the Author):

Garrido-Rodriguez et al., focuses on context-specific signaling pathway inference from phosphoproteomics datasets (online and in-house) and kinase-substrate networks. They revised EGFR signalling and proposed that literature-curated networks combined with network propagation yield the best recovery of signalling interactions. Finally, they claim that this analysis will help finding novel treatments against EGFR-driven diseases.

The manuscript is a novel addition to a series of emerging studies (listed and well referenced by the authors) that aim to challenge the status quo of the “traditional understanding of signaling pathways”. It expands on several fronts, most importantly, by 1) analysing short-term stimulated phosphoproteomes, 2) testing several computational methods, and 3) comparing the results to multiple well-defined ground truth sets.

However, a few concepts throughout the manuscript need clarification or more evidence especially for non-expert readers. The method described here have a great potential to be used by signalling experts and computational biologists but requires more qualitative examples addressed to a broad readership.

We thank the reviewer for their thoughtful summary of our work and agree that the manuscript could benefit from a clearer and more qualitative presentation of the resources, datasets, and methods introduced. In response, we have substantially revised the manuscript to enhance clarity and accessibility. Specifically, we have (A) added schematic illustrations and qualitative examples throughout to better convey the study’s scope and key insights; (B) removed results and metrics that might confuse a broader audience or lacked a clear message, such as direct versus indirect interactions, absolute kinase activity differences, and PageRank damping factor analysis, and rewrote many sections for more concise and direct communication; (C) incorporated two external datasets in non-EGF-stimulated contexts (kidney cells stimulated with transforming growth factor beta and HEK293 cells stimulated with interferon

alpha) to demonstrate broader applicability and provide a baseline for evaluation; and (D) consolidated and modularized all code into a reusable Python package to facilitate wider adoption and use in the phosphoproteomics field.

The idea of combining available information on kinase-substrate relationship and in-house generated datasets is modern and well executed. However, I have the following comments:

Major comments:

1. Results. From “To determine” on page 5 until “compared to literature ” on page 6 there are only two panels (Fig 1A and Fig S1A). A workflow is needed as well as some more illustrations on what the author did to come to this first conclusion.

We now provide a workflow illustrating how the kinase activity analysis was conducted using the input data from Hijazi et al., along with the various sources of prior knowledge and the two main variables considered when determining the appropriate threshold: data coverage and AUROC of common kinases. We have also included a heatmap to visually summarize the kinase activities from the Hijazi et al. dataset, aiming to make the analysis more accessible and visually clear. In addition, we have added new text to the Results section to better explain the methods used in these analyses.

[Figure Redacted]

“After obtaining the differential phosphoproteomic profiles from the kinase inhibitor treatments, we analyzed kinase activity by applying the Z-score method described in Müller-Dot et al. (Fig. 1b). This analysis used the three previously described resources, plus a combined resource containing the union of interactions from all three. We applied three thresholds for each resource (lenient, moderate, and strict), retrieving all interactions above these thresholds (see Methods). For each resulting network (defined by a specific resource and threshold), we calculated the proportion of phosphosites covered (Coverage) and the AUROC for kinase activities, using drug targets from the complementary in-vitro assay as true positives (Fig. 1a; see Methods).”

2. Results. “far exceeded the known kinase-substrate interactions from the literature”. This is not shown anywhere I can see. Please, provide the evidence for this sentence.

We apologize that this was not clearly explained before. We now explicitly point to the figure that compares the number of phosphoproteins, phosphosites, and interactions to those reported in the literature, making this point clearer and better supported.

“In a descriptive analysis of the expanded networks, we observed that the numbers of phosphosites, phosphoproteins, and interactions were far greater than those reported in the literature (Fig. 1d).”

3. Results: “kinases with similar evolutionary background do not share a high amount of targets”. Provide references and/or evidence.

We acknowledge that this claim was not properly supported by data in the first version of the manuscript and that the term ‘similar evolutionary background’ was confusing. We removed the ‘evolutionary’ term from our results and

changed the terminology of the analysis to ‘sequence similarity’ instead. To provide quantitative support to the claim, we performed an analysis of the target overlap coefficient for kinases sharing more than 50% sequence similarity. The results demonstrate that Phosformer outperforms other resources in target similarity among kinases with a comparable sequence similarity.

“We also examined the overlap for kinases with similar sequences. To do so, we calculated the overlap coefficient for target sites among kinases with >50% sequence similarity (N =1677) (Fig. 1f). This analysis revealed that kinases with similar sequences share a higher proportion of target sites in Phosformer (81%) compared to the kinase libraries (60%) or literature (26%), which makes sense, given that Phosformer relies solely on kinase sequences as input.”

4. Results: “As expected, adding new interactions reduced the frequency of specific targets, defined as phosphosites regulated by a single upstream kinase”. Again, this is not clear and not supported by evidence.

We apologize for the lack of clarity in the initial presentation of our results. We have revised the figure to include a small diagram that matches the definitions of “specialized” and “convergent” sites introduced in Johnson et al. We hope this claim is now clearer and better supported.

“We also examined specialized sites (targeted by ≤ 3 kinases) and convergent sites (targeted by ≥ 3 kinases), as defined by Johnson et al. This analysis mirrored the overall interaction counts: the literature-based network had the highest proportion of specialized sites (95%), followed by Phosformer (40%) and the kinase library (23%) (Fig. 1g).”

5. Results, beginning of page 7: “uncovering new interactions across various kinase families.” There are neither quantitative nor qualitative evidence.

We agree with the reviewer that the original analysis was not quantitative and relied primarily on the kinome tree figure (Figure 1E), which, while visually appealing, did not effectively convey quantitative differences between kinase subfamilies. To address this limitation, we have removed the kinome tree figure and instead analyzed the proportions of interactions by resource and by kinase superfamily (dual specificity, tyrosine kinase, and serine/threonine kinase). We believe this new figure provides a clearer and more quantitatively supported representation of our findings.

“We next examined how these interactions were distributed across major kinase families for each resource (Fig. 1e). In the literature-based network and kinase libraries, most interactions involved Serine/Threonine kinases (82% and 93%, respectively), whereas Phosphoformer had a higher proportion of interactions for Tyrosine kinases (62%). Dual-specificity kinases were underrepresented in all resources, comprising less than 5% of interactions.”

6. Results. Where do you show “state-of-the-art kinase-substrate interactome”?

We apologise if this was confusing. What we meant by “state-of-the-art kinase-substrate interactome” was to integrate state-of-the-art tools to expand the traditional kinase–substrate interactions typically used in downstream analyses. To clarify this now, instead of framing the integrated resource as a new state-of-the-art, we emphasize its value as an integrated resource, making it easily accessible and encouraging its use by the community for downstream applications. In addition, a new tutorial showing how to use the integrated resource for kinase activity analysis can now be found here:

https://github.com/saezlab/phosphonetworks/blob/main/analysis_tutorial.ipynb

“Overall, this resource integrates recent computational and experimental methods to greatly expand the number of kinase-substrate interactions. The four context-agnostic resources used in this study are accessible via a Python interface (see Methods), allowing researchers to easily examine how different prior knowledge networks influence downstream analyses.”

7. Results: how do they define canonical EGFR pathways? Please, clarify

In the original manuscript, we used the term “canonical” to describe the EGFR signaling pathway curated in the SIGNOR database, which contains interactions well supported by evidence. However, we recognize that calling a biological process “canonical” can be misleading. To avoid this ambiguity, we have replaced all instances of the “canonical pathway” terminology with “SIGNOR

EGFR signaling pathway” and now present its content explicitly in the new Figure 4C. This revision clarifies our intent and makes it clear exactly which pathway is being referenced.

"We drew on the curated SIGNOR EGFR signaling pathway, which provides well-established, experimentally validated interactions. From this pathway we extracted only kinase–kinase links, yielding 23 high-confidence interactions (Fig. 4c)"

8. Results: after presenting the workflow in Figure 4 the authors only describe the different steps of their analysis and the comparison between datasets and type of data. What is missing are examples at each step or, in other terms, a qualitative evaluation of their main conclusions. Which kinase-substrate are they talking about? This would enormously help the reader (and ultimately, the users of their methods) to apply these methods to other datasets

We agree that the presentation of the results was convoluted and lacked clarity, as also noted by Reviewer #3. In response, we have created entirely new versions of Figures 4 and 5, removed metrics and results that did not contribute to a clear message, and provided a more qualitative, illustrative breakdown of each part of the network analysis. We believe these changes make the revised section clearer, more coherent, and easier to read. Because this revision required rewriting two entire sections of the paper, we are not pasting here the updated text, and kindly invite the reviewer to examine it in the revised manuscript.

9. Figure 2D-E. The results should be discussed in more detail, as the large differences/variance among different studies is critical for the interpretation of later results.

In response to this comment, we performed a more detailed analysis of correlations between the different studies at the phosphosite level. The revised Figure 2 highlights these differences more clearly by bringing the volcano plots and correlation analyses to the forefront. We now explicitly acknowledge in the manuscript that overall agreement across studies is low when considering all sites, but that the correlation increases when focusing on significantly changing sites. To further support this discussion, the new Figure 5 presents the network evaluation results at the study level, allowing readers to more easily explore study-specific effects. We believe these additions provide a deeper and clearer discussion of the observed differences and variance among studies.

“The studies included in the meta-analysis showed heterogeneity in both the magnitude and significance of phosphoproteomic changes following EGF stimulation (Fig. 2b). In particular, the three DIA HeLa studies (Bortel, Lancaster, and Skowronek) produced broader volcano plots, reflecting stronger overall changes in magnitude compared to the

TMT-DDA HEK293 datasets generated in this study, as well as the Tuechler (TMT-DDA) and Chen (SILAC-DDA) datasets. These differences likely reflect the difference in proteomic technologies employed but could also be explained by the underlying biological context, since HeLa and HEK293 cells provide distinct cellular backgrounds. Despite these differences, most phosphosites belonging to the EGFR signaling pathway, as defined in SIGNOR, were consistently up-regulated across all datasets (Fig. 2c). This observation underscores that while experimental platforms and cell types may influence the extent of the detected changes, the core EGF response remained reproducible across studies.”

[...]

“To evaluate agreement between studies beyond EGF pathway sites, we compared all quantified sites using both overlap and correlation measures (Fig. 2d). Jaccard similarity coefficients showed only moderate overlap (0.07–0.43), with the highest value observed between two runs on the same cell line (HEK293F). Spearman correlation scores of fold-changes were also modest (0.01–0.41), and the average correlation between the control studies and the EGF datasets was lower (0.08) than that within the EGF datasets (0.29), indicating greater reproducibility of the EGF signal. When the analysis was restricted to significantly changing phosphosites, reproducibility improved substantially: Spearman’s rho values rose to 0.33–0.71, with the strongest agreement seen between the Skowronek and Bortel datasets. No overlapping significantly regulated sites were found for control datasets (Chen and Tuechler) in this analysis. Next, exploiting the time resolution of the HEK293F datasets, we compared log fold-changes of significant sites across time points (Fig. 2e). Here, correlations were higher at early time points (0.64–0.73 up to 15 minutes) than at the later 25-minute time point within the first experiment (X-axis). For the second study, correlation with the first experiment increased with post-stimulation time, from 0.58 at 1 minute to 0.7 at 15 minutes (Y-axis). Together, these results suggest that although overall overlap and agreement are moderate, reproducibility improves when focusing on significantly regulated phosphosites and is affected by the time of measurement after initial stimulation with EGF.”

10. Figure S3C. The difference between the 3 different network inference algorithms would be easier to understand if Supplementary Figure S3C was included in the main text along with the explanation of differences in topology. This would also make this part more accessible to a more general audience.

To make the network component of the paper more accessible and easier to understand, we now include illustrative kinase–kinase networks generated by applying the three methods with the following configuration: number of interactions = 10, data = this study (HEK293T), and resource = literature. We believe this new figure, together with our revised text describing the methodological principles of each approach, makes the differences between methods clearer and easier to understand to a more general audience.

“Regarding methods to infer kinase–kinase pathways, we developed three ad-hoc strategies, each of them using the underlying principles of published approaches (Fig. 4b; subnetworks obtained with the three methods using ‘literature’ as a resource, number of interactions = 10, and data from this study). The first method, inspired by CausalPath27, scores each interaction by the average change in kinase activity of its source and target (“Source/Target mean activity”) and generates a subnetwork from the top-scoring N interactions; this simple baseline uses kinase-level data within a kinase–kinase network but does not incorporate network topology or enforce a directed acyclic graph (DAG). The second method employs network propagation similar to TieDIE28 and phueGO29, running a personalized PageRank analysis and selecting top interactions based on the mean PageRank scores of their source and target nodes (“Source/Target Personalized PageRank score”, PPR); this approach accounts for network topology and can include

kinases without activity estimates if their PageRank scores are high enough, yet it still does not enforce a DAG. The third method applies a “Rooted prize-collecting Steiner tree” (RPCST) formulated as an Integer Linear Programming problem, inspired by TPS30 and PHONEMES31, and unlike the others it guarantees a fully connected subnetwork and enforces a DAG structure, prohibiting feedback loops when given an input root node (here, EGFR). The three methods require three inputs: a kinase–kinase prior network, a list of kinase activities (such as those derived in the previous section), and a specified number of edges. This edge parameter determines how many top interactions are selected in the first two methods and serves as a strict constraint in the RPCST approach, ensuring that all three strategies generate subnetworks of comparable size.”

11. Discussion: how would the authors envision to apply their methods to signalling pathways less known or understudied compared to the EGFR pathway and in more relevant cell models or primary cells? They present limitations of their study, but these points require more discussion

We agree that adding this perspective will benefit the manuscript’s readers, and it was also highlighted by the other reviewer. We have therefore added a new paragraph to the discussion that presents this viewpoint and addresses its main challenge, defining ground-truth interactions:

“The main challenge in this work was defining a reliable ground truth for kinase–kinase directed interactions. We could partially address it because our study focused on a well-characterized signaling process with a commonly used stimulus and abundant data. Even within the EGF context, however, there is no universally accepted pathway: major knowledge bases such as SIGNOR³⁵, Reactome³⁶, and WikiPathways³⁷ differ in their representations of this signaling cascade. Defining a ground truth would be even more challenging in less-studied stimuli or biological contexts, such as primary cells. In those cases, one option is to pair condition-specific experiments with a broad set of perturbational studies in the same system to identify correlation-supported interactions more reliably, as we did for our third ground truth set. Such data could even support the creation of a context-specific, data-driven prior network to replace the kinase–substrate context-agnostic resources used here. Another strategy is to begin with kinase activity analysis in response to a single perturbation, identify modules of highly deregulated

kinases using the workflow presented in this manuscript, and then validate the inferred interactions through targeted, small-scale perturbational experiments, such as kinase inhibition combined with the relevant stimulus.”

12. General comment: Why did the authors choose to use different time points of EGF stimulation? It is well-known that (among others) the duration of stimulation drives different cellular responses. This makes the comparison of these time-resolved datasets to literature datasets with a single time point problematic, which is not addressed in the manuscript.

We selected different time points because, based on classic literature and the prevailing view of the EGF pathway, we expected to observe major clusters of phosphosites showing oscillatory behavior—for example, sites upregulated at an early time point and downregulated later. We hypothesized that this pattern would emerge with higher temporal resolution, which led us to perform a second time-resolved experiment in HEK293F cells. However, as now shown explicitly in the manuscript, we did not see such oscillatory behaviour. Silhouette analysis across multiple clustering options revealed only two main clusters: sites upregulated in response to the stimulus and sites downregulated. This finding also motivated the use of our experiment with higher time-resolution to define a third ground truth set (namely, correlation). The reviewer is correct, however, that comparisons could be problematic toward datasets with more time points relative to those with a single time point. To address this, in the revised manuscript we retain only the time point from each time-resolved experiment that shows the largest number of changes, defined as the highest average absolute log-fold change.

“To explore whether EGF stimulation produces consistent complex temporal patterns in our datasets (such as oscillations driven by feedback mechanisms), we clustered log-fold changes from phosphosites that showed significant differences at any time point (adjusted $P \leq 0.05$ and $|\logFC| > 1$) compared to unstimulated cells. Silhouette analysis of K-means clustering across different values of K indicated that the best separation was consistently achieved with $K = 2$ (Supplementary Fig. 2a). This suggests that most EGF-induced changes fall into two main groups over time: up-regulated and down-regulated sites (Supplementary Fig. 2b). For downstream comparisons across studies, we selected, for each of our studies, the time point with the largest deviation from unstimulated cells, defined as the maximum absolute average log-fold change across all phosphosites (9 min for HEK293T, 3 min for HEK293F, and 12 min for HEK293F TR).”

13. General comment. For the generalizability of the results, the inclusion of only two cell lines is not enough, however, it makes the results harder to interpret. Did the authors test whether only including data from HEK293 cells (data from this study) or data from HeLa cells (literature data) changes the results/optimal method?

We agree with the reviewer that analyzing only two cell lines and a single stimulus limited the generalizability of our findings. To expand the scope, we incorporated an external dataset of kidney cells stimulated with transforming growth factor beta (TGF β). In addition, all results in the main figures are now shown separately for each study, and we found no notable differences between the HeLa and HEK293 datasets in the evaluation of kinase–kinase signaling pathway inference. As noted in the revised manuscript, however, the three HeLa datasets were generated using DIA phosphoproteomics, whereas the data from this study were produced with DDA–TMT–labeled phosphoproteomics, making it difficult to attribute any differences to a biological effect.

“The studies included in the meta-analysis showed heterogeneity in both the magnitude and significance of phosphoproteomic changes following EGF stimulation (Fig. 2b). In particular, the three DIA HeLa studies (Bortel, Lancaster, and Skowronek) produced broader volcano plots, reflecting stronger overall changes in magnitude compared to the TMT-DDA HEK293 datasets generated in this study, as well as the Tuechler

(TMT-DDA) and Chen (SILAC-DDA) datasets. These differences likely reflect the difference in proteomic technologies employed but could also be explained by the underlying biological context, since HeLa and HEK293 cells provide distinct cellular backgrounds. Despite these differences, most phosphosites belonging to the EGFR signaling pathway, as defined in SIGNOR, were consistently up-regulated across all datasets (Fig. 2c). This observation underscores that while experimental platforms and cell types may influence the extent of the detected changes, the core EGF response remained reproducible across studies.”

Minor comments

1. Introduction. The sentence “Such an analysis would allow us to quantify how our traditional view of signaling pathways aligns with the broader patterns of activation revealed by new methods with larger coverage” require clarification and contextualization.

We have now extended this sentence to be more clear and contextualized:

“Such an analysis would make it possible to directly compare established, textbook descriptions of signaling pathways with the activation patterns uncovered by modern high-coverage phosphoproteomic methods. This comparison would not only quantify how well current context-specific data support traditional pathway models but also reveal additional connections or alternative signaling routes that extend or refine our understanding. Moreover, it would help differentiate the contributions of context-specific data and the choice of prior-knowledge resources.”

2. Introduction. EGFR signalling is crucial in breast, lung, melanoma and at least also glioblastoma.

These two cancer types are now added on every mention to specific tumors throughout the text.

3. Results: in the sentence, "This approach allowed us to generalize known interactions from literature to kinases and substrates with similar evolutionary backgrounds." It is unclear what "similar evolutionary background" refers to.

We have now clarified what this means within the main text and in the response to the Major point 4.

"Phosformer's greater coverage of Tyrosine kinases may result from biases in its training data or from similar predictions for kinases of similar sequence. To quantify this, we calculated the overlap coefficient for target sites among kinases with high sequence similarity (>50% sequence similarity, N =1677)."

4. Results: the three Excel files submitted as supp material are not cited in the text. It is not clear what is shown in each of them

We agree with the reviewer that including these supplementary files in the initial submission—without documentation or references in the main text—was not the most effective way to share the intermediate tables generated by our analyses.

To improve accessibility, we now provide all resources, data tables, and network methods as part of the code accompanying the manuscript, available at:

- <https://github.com/saezlab/phosponetworks>

The relevant tables can now be retrieved directly within any Python environment using the following functions:

- `pp.kinsub.get_combined_kinsub()`
 - Retrieves all combined kinase–substrate interactions across thresholds.
- `pp.phosphodata.get_all_studies()`
 - Retrieves all data gathered during the meta-analysis.
- `pp.methods.egf_kinase_activity_analysis()`
 - Uses the two tables above to compute kinase activities and return the results.

5. The results on Figure S2C should be included in the main text and discussed

We agree with the reviewer that these results are important. That said, we found that the absolute kinase activity difference metric was too complex to present in a way that would be clear to a non-computational audience. Instead, we now highlight this aspect through the new main Figure 2e and Supplementary Figure 2, which provide a more accessible, detailed view of the time-resolution analysis that was previously buried in Supplementary Figure 2C.

Reviewer #3 (Remarks to the Author):

In this manuscript, Garrido-Rodriguez, Potel and colleagues explore different ways to infer pathways from large-scale phosphoproteomics data, using both new phosphoproteomic data and existing datasets. They compare different pathway inference methods and different ground truth sets to evaluate the performance of pathway inference, focusing on the well-characterized EGF/EGFR pathway. The topic is certainly important, since we are being inundated with large-scale datasets from different sources, and making biological sense of these datasets is of utmost importance. At the same time, it is a big service to the community to evaluate different methods and suggest gold standard approaches for analyzing large-scale (phosphoproteomics) data.

However, after reading this manuscript as a biologist who is interested in signaling networks and interactions, I am not sure how much I have learnt. To be clear, I cannot comment on the methodological aspects or the computational analysis (although I have some general questions/suggestions below), but as I was trying to distill the results of the manuscript to an actionable output or to a fundamentally novel insight, I had hard time doing so. Perhaps this is due to the writing style that is not very accessible to an experimentalist and lists results in a matter-of-fact manner without highlighting the most relevant insights. Overall, I am not quite sure if this manuscript moves the needle enough to warrant publication in its current form in Nature Communications.

We thank the reviewer for their thoughtful assessment of our work and for recognizing the significance of the research. We also agree that several sections of the manuscript required clearer presentation. In response to this, we undertook a substantial revision that includes the following major improvements: (A) added schematic illustrations and qualitative examples throughout to better convey the study's scope and key insights; (B) removed results and metrics that might confuse a broader audience or lacked a clear message, such as direct versus indirect interactions, absolute kinase activity differences, and PageRank damping factor analysis, and rewrote many sections

for more concise and direct communication; (C) incorporated two external datasets in non-EGF-stimulated contexts (kidney cells stimulated with transforming growth factor beta and HEK293 cells stimulated with interferon alpha) to demonstrate broader applicability and provide a baseline for evaluation; and (D) consolidated and modularized all code into a reusable Python package to facilitate wider adoption and use in the phosphoproteomics field.

Conceptually, our central goal is to illuminate the broader landscape of signaling events that extend beyond “traditional” pathway definitions, integrating both data-driven and knowledge-based perspectives. While this approach may not always yield immediate concrete biological conclusions, we believe that identifying and quantifying these previously uncharacterized signaling mechanisms—even within a well-studied pathway such as EGF—has the potential to be transformative, opening new directions for both large-scale proteomics and focused, mechanistic investigations of signaling processes.

General comments

1. It would be useful to include full ROC curves in the supplement, as the differences between the different resources are relatively modest by just looking at AUROC (~0.60 to ~0.68 in Fig 1A)

In response to this comment, we have also provided the full AUROC values in the supplementary materials. We agree with the reviewer that the gain in accuracy is modest, and we have added text to explicitly highlight this for the reader.

“Although differences in predictive performance between resources were modest, the best score came from Phosformer under the lenient threshold (AUROC=0.67, Fig. 1c, Supplementary Fig. 1a).”

2. It is intriguing that Phosformer is so highly enriched for tyrosine kinases (Fig 1E), which are arguably the best-characterized kinases due to their importance in tumorigenesis and the availability of many specific inhibitors. I wonder how much this difference contributes to differential performance of different kinase-substrate networks. For example, does Phoshformer perform better (or worse) if the analysis is limited to tyrosine kinases?

We agree with the reviewer that Phosformer appears to better capture Tyrosine kinases compared to Serine/Threonine kinases. We also acknowledge that this may stem from inherent research biases or the overlap in target sets among similar kinases. Unfortunately, we were unable to perform an independent benchmark for the tyrosine kinase family using the Hijazi dataset, as the number of commonly shared true positives and negatives was too small to support meaningful per-family analyses. For reference, we could only estimate the activity for two tyrosine kinases using literature as a resource: ABL1 and SRC. The Hijazi dataset contains 122 Tyrosine phosphorylation sites, compared with 12,059 Serine and 2,694 Threonine phosphorylation sites. Nevertheless, we now explicitly highlight this bias toward Tyrosine kinases in Figure 1E and mention potential explanations for this in the results.

E
“We next examined how these interactions were distributed across major kinase families for each resource (Fig. 1e). In the literature-based network and kinase libraries, most interactions involved Serine/Threonine kinases (82% and 93%, respectively), whereas Phosformer had a higher proportion of interactions for Tyrosine kinases (62%). Dual-specificity kinases were underrepresented in all resources, comprising less than 5% of interactions. Phosformer’s greater coverage of Tyrosine kinases may result from biases in its training data, or from a better performance of the language model in this kinase subfamily.”

3. The authors use three previously published EGF phosphoproteomics datasets in HeLa cells and generate three other datasets in HEK293 cells. It is not clear why HEK293 cells were selected as a model (they express very low amounts of EGFR) in contrast to other cell lines that have much higher EGFR expression.

We agree with the reviewer that there could be arguments to choosing other cells instead of HEK293 cells; as the reviewer points out, other cell lines express higher amounts of EGFR. For the purpose of our study, it was important to use a suspension-adapted cell line. We hypothesized that ligand accessibility might differ between cells grown in suspension and those grown as an attached monolayer, making the floating version more suitable for our time-course experiments. We chose for this reason the suspension-adapted HEK293F variant over other cell lines that would have higher levels of EGFR, but would not grow in suspension.

4. It would be useful to add another phosphoproteomic dataset or datasets as a control for specificity. For example, what would the Jaccard similarity look like if similar datasets for other RTKs or e.g. TGFbeta signaling were included as outgroups/controls?

We agree with the reviewer that including datasets with different stimuli or biological contexts would help test the generalizability of our results and provide a baseline for comparing kinase activity network evaluations. In response, we added two external datasets in non-EGF-stimulated contexts: one from kidney cells stimulated with transforming growth factor beta (TGF β) and another from HEK293 cells stimulated with interferon alpha (IFN α). These datasets demonstrate how our methods can be applied to alternative contexts and serve as reference points for our evaluations.

The addition of these datasets yielded new insights at both the kinase activity and network analysis levels. These results, now presented in the revised and clearer Figure 3 and Figure 5, also highlight how prior knowledge influences the recovery of EGF-related interactions regardless of the underlying data. In the original manuscript, we illustrated this effect using the variable damping factor of the PageRank algorithm; we have now replaced that approach with a more straightforward explanation, showing that EGF interactions are recovered even in non-EGF studies, which provides a clearer explanation. The updated manuscript details these findings at both the kinase level and in the network analysis evaluation.

“To explore kinase activity changes, we integrated the meta-analysis data with the kinase-substrate resources and performed kinase activity analysis using differential phosphosite data as input. We also assessed data coverage and compared studies at the kinase activity level (Fig. 3a). Coverage patterns were consistent with Hijazi et al., with the combined resource providing the broadest coverage across studies (70%), followed by the kinase library (64%), Phosformer (21%), and literature (4%). Importantly, for all resources, average correlations of kinase activities were higher between EGF studies than between EGF and control studies (mean Spearman’s Rho = 0.65 vs. -0.07, respectively).

This indicates that despite variability in coverage, kinase-level agreement across EGF studies remained robust.”

[Figure Redacted]

[...] “Differences between EGF-stimulated and control datasets were minimal, with only the Phosformer resource paired with the mean activity method showing a small distinction: control studies recovered one interaction, whereas EGF stimulation yielded up to four. [...]

[...] “Differences between control and EGF conditions were again minor, rarely exceeding three interactions. The limited overlap between this ground truth and the kinase–kinase networks likely explains the overall low recovery.” [...]

[...] “EGF stimulation studies produced slightly higher overlaps than controls, but differences remained minimal (average overlap difference = 3.6 in the best settings: Phosformer as resource, Source/Target mean as method and Correlation (Lenient) as ground truth set).“ [...]

[...] “Across all ground truths, the prior network determined recovery far more than experimental context or reconstruction method. Cell type and stimulation had little effect, and the modest gains from PPR or mean activity compared with RPCST were minor in comparison to the strong advantage of the Literature prior.” [...]

5. Did the Jaccard similarity calculation only consider phosphopeptides that were detected in both studies or did it consider all phosphopeptides? The former would be a better measure of the similarity of the biological responses,

since mass spec instruments and experimental conditions affect the sensitivity of each assay differently.

We thank the reviewer for highlighting that different overlap metrics capture different aspects of similarity and deserve closer examination. The Jaccard similarity considers all identified sites. To expand this, the revised manuscript now reports three complementary measures in the main figure: (1) Jaccard index of all detected sites, (2) Spearman correlation of log-fold changes across all sites, and (3) Spearman correlation of log-fold changes limited to significantly deregulated sites in each study. These additional analyses show that while overall agreement remains low, focusing on significantly deregulated sites substantially increases reproducibility across experimental conditions.

D

“To evaluate agreement between studies beyond EGF pathway sites, we compared all quantified sites using both overlap and correlation measures (Fig. 2d). Jaccard similarity coefficients showed only moderate overlap (0.07–0.43), with the highest value observed between two runs on the same cell line (HEK293F). Spearman correlation scores of fold-changes were also modest (0.01–0.41), and the average correlation between the control studies and the EGF datasets was lower (0.08) than that within the EGF datasets (0.29), indicating greater reproducibility of the EGF signal. When the analysis was restricted to significantly changing phosphosites, reproducibility improved substantially: Spearman’s rho values rose to 0.33–0.71, with the strongest agreement seen between the Skowronek and Bortel datasets. No overlapping significantly regulated sites were found for control datasets (Chen and Tuechler) in this analysis. Next, exploiting the time resolution of the HEK293F datasets, we compared log fold-changes of significant sites

across time points (Fig. 2e). Here, correlations were higher at early time points (0.64–0.73 up to 15 minutes) than at the later 25-minute time point within the first experiment (X-axis). For the second study, correlation with the first experiment increased with post-stimulation time, from 0.58 at 1 minute to 0.7 at 15 minutes (Y-axis). Together, these results suggest that although overall overlap and agreement are moderate, reproducibility improves when focusing on significantly regulated phosphosites and is affected by the time of measurement after initial stimulation with EGF.”

6. What is absolute difference in kinase activity (page 12 and Fig 3C)? This is poorly described in the results section and therefore interpretation of the results is quite difficult.

We agree with the reviewer that the “absolute difference in kinase activity” metric was unclear and have removed it from the manuscript to improve accessibility for a broader audience. Our original goal was to measure how strongly kinases in the EGF SIGNOR pathway responded compared with other kinases, but the metric was confusing both because it represented absolute changes (without indicating up- or down-regulation) and because the results were not broken down at the individual kinase level. To address this, we now present a clearer Figure 3b showing the kinase activity changes for the 25 most strongly regulated kinases in response to EGF across studies and resources, allowing readers to interpret these differences directly.

“We next assessed which individual kinases showed the strongest activity changes across different resources used for the EGF studies (Fig. 3b). The top 25 regulated kinases varied considerably depending on the resource. EGFR, the main receptor kinase activated in response to EGF, was only among the top hits when using the literature-derived resource. By contrast, the Kinase Library primarily highlighted members of the RSK (RPS6KA1–6) and MAPKAPK families (MAPKAPK2/3/5), along with MAPK7, all established downstream effectors of MAPK signaling. Predictions from Phosformer were more diverse, emphasizing SGK family members (SGK1–3), PKA isoforms (PRKACA, PRKACB, PRKACG), an RSK isoform (RPS6KA5), and PKG1 (PRKG1). When combining all resources, the results were again dominated by the RSK family (RPS6KA1–6), underscoring the consistency of these effector modules across methods. In comparison, the literature-based set retained a more established profile centered on EGFR, MAPK1/3, and their classical partners RAF1, INSR, and Src-family kinases (LCK, FYN), as well as RPS6KB1 and PRKCD. Notably, several kinases of the pathway appeared across top hits on multiple resources, including MAPK1, MAPK3, and AKT isoforms (AKT1, AKT2, and AKT3).”

7. The authors mention that in the HEK293F-TR experiment, the peak of activation (absolute difference in kinase activity?) shows two peaks, which was absent in non-time resolved studies. Looking at the data in Fig S2C, the activity in HEK293F (middle panel) seems to show an opposite trend compared to HEK293F-TR (left panel). In the HEK293F experiment, 3-min timepoint shows lower activity than the 9-min timepoint, whereas in the HEK293F-TR

experiment, the activity is higher at 3 minutes and lower in 9 minutes. If the 3-min and 9-min experiments show such a different pattern although they should (to my understanding) be essentially identically performed, how confident can one be about these patterns?

We acknowledge that the original presentation of these results was confusing. As noted in our previous response, the “absolute kinase activity difference” metric combined up- and down-regulated changes, making it difficult to interpret and potentially obscuring time-dependent patterns. To address these issues, we have revised the manuscript and added two new figures that explore the dynamic component of our time-resolved data more deeply.

“To explore whether EGF stimulation produces consistent complex temporal patterns in our datasets (such as oscillations driven by feedback mechanisms), we clustered log-fold changes from phosphosites that showed significant differences at any time point (adjusted $P \leq 0.05$ and $|\logFC| > 1$) compared to unstimulated cells. Silhouette analysis of K-means clustering across different values of K indicated that the best separation was consistently achieved with $K = 2$ (Supplementary Fig. 2a). This suggests that most EGF-induced changes fall into two main groups over time: up-regulated and down-regulated sites (Supplementary Fig. 2b). For downstream comparisons across studies, we selected, for each of our studies, the time point with the largest deviation from unstimulated cells, defined as the maximum absolute average log-fold change across all phosphosites (9 min for HEK293T, 3 min for HEK293F, and 12 min for HEK293F TR).”

Regarding the reviewer’s second point about the similarity between the two HEK293F datasets, we agree that a stronger correlation over time would indeed be expected. We believe, however, that the observed correlation values between

the two studies can be attributed to several factors. Variations in cell density, growth conditions, or the number of passages in the cell line could explain differences in the temporal response to a growth stimulus. Moreover, phosphoproteomics, and post-translational modification measurements in general, are known to be more sensitive than protein abundance readouts, which are inherently more difficult to modulate.

That said, a deeper analysis of the correlation among significantly affected phosphosites over time revealed an important pattern. Specifically, the first two time points of the initial experiment (3 and 9 minutes) show higher correlation with all time points of the second experiment (up to 15 minutes) compared with the 25-minute time point. This suggests that, while the overlap is not perfect, our experiments demonstrate reproducibility in their temporal response.

“Next, exploiting the time resolution of the HEK293F datasets, we compared log fold-changes of significant sites across time points (Fig. 2e). Here, correlations were higher at early time points (0.64–0.73 up to 15 minutes) than at the later 25-minute time point within the first experiment (X-axis). For the second study, correlation with the first experiment increased with post-stimulation time, from 0.58 at 1 minute to 0.7 at 15 minutes (Y-axis). Together, these results suggest that although overall overlap and agreement are moderate, reproducibility improves when focusing on significantly

regulated phosphosites and is affected by the time of measurement after initial stimulation with EGF.”

8. The kinase activity analysis in Fig 3D: I'm a bit puzzled about the results and interpretation. If the cells are stimulated with EGF, shouldn't EGFR be the top kinase in any analysis that is biologically relevant? The fact that only one method (literature) actually captures what is happening at the cells is alarming. Although it is certainly true that many other kinases are activated downstream of EGFR, it is biologically unlikely that they would be activated more in terms of fold-changes in target phosphorylation and/or in terms of total fraction of targets regulated. As an end user, one would hope that in this extremely well controlled situation where only a single ligand is added, the primary target of the signaling cascade would light up in the analysis. If this is not the case, what hope do we have for all the murky biology that happens in more natural contexts? I would really appreciate if the authors commented and discussed this.

We appreciate the reviewer's observation, which touches on one of the most central points of our manuscript. When literature-derived prior knowledge is used as a resource, EGFR emerges as the top kinase, an outcome that is expected, as the reviewer points out. At the same time, the other resources we tested also highlight kinases that are well known to participate in the EGF response, even if EGFR itself is not ranked first. This highlights the value of using different resources for kinase activity estimation as proposed in the manuscript.

What our analysis cannot definitively answer is whether these downstream kinases exhibit stronger activation than the upstream receptor. As per the kinase activity analysis, many of them have a larger number of significantly regulated downstream phosphosites than EGFR, which is directly responsible for their prominence in the results. One possible explanation is that the larger abundance of downstream targets can amplify the apparent magnitude of activation. Another is that signaling networks behave as complex, dynamic systems, where a single well-defined stimulus can produce oscillatory or cascading effects that intensify the downstream response, similar to the

behavior of a double pendulum. We now explicitly discuss these possibilities in two sections of the revised manuscript: within the kinase activity analysis section and in the discussion.

“Together, these comparisons reveal two complementary pictures of the EGF signaling response. Literature-derived networks recover well-established kinases, consistent with decades of small-scale studies. In contrast, expanded resources such as Kinase Library and Phosformer point to broader sets of MAPK effector kinases, AGC kinases, and related families that could plausibly play important roles in shaping the downstream signaling landscape. While it remains difficult to validate or reject these expanded views, given that phosphoproteomics data never achieved this level of coverage, it is conceivable that the magnitude of activity changes in these kinases is comparable to those usually associated with EGF response. Thus, the broader resources may be revealing layers of the signaling response that extend beyond the traditional view of the pathway.”

[...]

“The main challenge in this work was defining a reliable ground truth for kinase–kinase directed interactions. We could partially address it because our study focused on a well-characterized signaling process with a commonly used stimulus and abundant data. Even within the EGF context, however, there is no universally accepted pathway: major knowledge bases such as SIGNOR³⁵, Reactome³⁶, and WikiPathways³⁷ differ in their representations of this signaling cascade. Defining a ground truth would be even more challenging in less-studied stimuli or biological contexts, such as primary cells. In those cases, one option is to pair condition-specific experiments with a broad set of perturbational studies in the same system to identify correlation-supported interactions more reliably, as we did for our third ground truth set. Such data could even support the creation of a context-specific, data-driven prior network to replace the kinase–substrate context-agnostic resources used here. Another strategy is to begin with kinase activity analysis in response to a single perturbation, identify modules of highly deregulated kinases using the workflow presented in this manuscript, and then validate the inferred interactions through targeted, small-scale perturbational experiments, such as kinase inhibition combined with the relevant stimulus.”

9. Finally, it would be useful to know how sensitive the authors’ analysis is to the selected cutoffs (which are explained in the methods section) in the used

datasets such as Hijazi et al. and the ground truth sets (e.g. kinase overexpression).

We recognize that our analyses could be influenced by the cutoffs and thresholds used to define the ground truth sets, including those for kinase overexpression and correlation. To assess this, we have now tested multiple threshold values for each benchmark. The kinase activity analysis proved robust, showing consistent results across different cutoffs. In contrast, the network benchmark was more sensitive, with the patterns observed in our initial analysis becoming more pronounced as thresholds were varied. These findings are now summarized in the revised manuscript and presented in the updated main Figures 4 and 5.

“Similar results were obtained when changing the inhibition cutoff that we used to consider a kinase a drug-target (Supplementary Fig. 1b)”

“Literature also dominated the overexpression ground truth. Only when applying a moderate threshold did Phosformer slightly exceed it, and then by a single interaction. Across methods, PPR recovered more interactions on average (3.6) than mean activity (2.4) or RPCST (2.3). Differences between control and EGF conditions were again

minor, rarely exceeding three interactions. The limited overlap between this ground truth and the kinase–kinase networks likely explains the overall low recovery.

Correlation-based ground truths (lenient, moderate, and strict) reinforced the same trend. Under the lenient threshold, Literature subnetworks recovered up to 29 interactions—more than double any other resource. Phosformer occasionally approached similar performance in individual studies, while kinase library and combined networks remained consistently low. EGF stimulation studies produced slightly higher overlaps than controls, but differences remained minimal (average overlap difference = 3.6 in the best settings: Phosformer as resource, Source/Target mean as method and Correlation (Lenient) as ground truth set). PPR provided a modest but consistent advantage (average overlap = 9.97, compared to 6.70 from Source/Target mean and 5.34 from RPCST), though differences between methods were far smaller than those between resources.”

[Figure Redacted]

Minor comments

10. The authors often refer to “deregulation” of signaling when analyzing EGF-induced changes. Although this is semantics, isn’t this just “regulation” since there is nothing abnormal about EGF inducing phosphorylation changes via EGFR?

We agree with the reviewer on this and have changed the terminology through the manuscript.

11. Fig S4B (page 16) should be Fig S5B

These supplementary figures have now been removed from the manuscript.